# The effects of quality of evidence communication on perception of public health information about COVID-19: Two randomised controlled trials

**Claudia R. Schneider** [1,2]*, **Alexandra L. J. Freeman** [1], **David Spiegelhalter**[1], **Sander van der Linden**[1,2]

**1** Winton Centre for Risk and Evidence Communication, University of Cambridge, Cambridge, United Kingdom, **2** Department of Psychology, University of Cambridge, Cambridge, United Kingdom

* cs2025@cam.ac.uk

**Data Availability Statement:** All data collected for the reported studies (de-identified participant data), along with the questionnaires used, is publicly available at https://osf.io/z6ps9/.

## Abstract

### Background

The quality of evidence about the effectiveness of non-pharmaceutical health interventions is often low, but little is known about the effects of communicating indications of evidence quality to the public.

### Methods

In two blinded, randomised, controlled, online experiments, US participants (total n = 2140) were shown one of several versions of an infographic illustrating the effectiveness of eye protection in reducing COVID-19 transmission. Their trust in the information, understanding, feelings of effectiveness of eye protection, and the likelihood of them adopting it were measured.

### Findings

Compared to those given no quality cues, participants who were told the quality of the evidence on eye protection was 'low', rated the evidence less trustworthy (p = .001, d = 0.25), and rated it as subjectively less effective (p = .018, d = 0.19). The same effects emerged compared to those who were told the quality of the evidence was 'high', and in one of the two studies, those shown 'low' quality of evidence said they were less likely to use eye protection (p = .005, d = 0.18). Participants who were told the quality of the evidence was 'high' showed no statistically significant differences on these measures compared to those given no information about evidence quality.

### Conclusions

Without quality of evidence cues, participants responded to the evidence about the public health intervention as if it was high quality and this affected their subjective perceptions of its efficacy and trust in the provided information. This raises the ethical dilemma of weighing

**Funding:** This project was carried out using core funding from The Winton Centre for Risk & Evidence Communication, which comprises a donation from the David & Claudia Harding Foundation.

**Competing interests:** The authors have declared that no competing interests exist.

the importance of transparently stating when the evidence base is actually low quality against evidence that providing such information can decrease trust, perception of intervention efficacy, and likelihood of adopting it.

## Introduction

In times of public health crises, such as a global pandemic, governments and individuals around the world are faced with the task of finding and taking effective measures of limiting the spread and impacts of the disease. On one end of the spectrum stand mandated measures that aim to prevent cross-human contact, such as border control actions, national lockdowns, and economic shutdowns [1–5]. On the other end stand measures which allow for human contact but trying to mitigate its impact, such as non-pharmaceutical interventions, which can be either mandated or left to individual choice. Non-pharmaceutical, behavioural interventions such as wearing face coverings, eye protection, and practicing social distancing [6, 7] are a vital part of the mitigations individuals can adopt to lower their risk of infection or transmission of viruses such as SARS-Cov-2. When effective vaccines are unavailable, not taken up or if additional protection is needed, such interventions are the only actions that individuals can take themselves [8–10] and they have been described as crucial for successfully managing the COVID-19 pandemic and reducing transmissions if implemented effectively [11–16]. Decisions about the adoption of such interventions, by policy-makers or individuals, are multi-faceted and can include political, economic, situational, personal health, psychological and other considerations [17–24], but also—importantly—information about evidence for their effectiveness [25–28]. Communication about such interventions, including their effectiveness, to both the public and policy-makers is therefore crucial.

However, despite observational data showing the overall effectiveness of suites of interventions in countries that have mandated various behaviours to manage the pandemic [11, 12, 14, 16], the evidence around the effectiveness of each potential non-pharmaceutical intervention is still emerging. Attempts at quantification of their effectiveness (e.g. How much does wearing eye protection reduce the chance of infection or transmission of COVID-19?) leads to a number of levels of uncertainty. Any experimental or observational data can give a point estimate (e.g. a percentage point reduction in the chance of infection or transmission) with a confidence interval ('direct' uncertainty, as defined in [29]). Meta-analyses can combine such estimated ranges, but the quantified uncertainty in confidence intervals only reflects a certain amount of the total uncertainty at play. Systematic biases, such as stemming from shortcomings in study design or data collection processes, unexplored variation and a host of other factors that are not easily quantified and cannot be deduced from the effectiveness estimate and its confidence interval, cause more 'indirect' uncertainties. These 'indirect' uncertainties—not directly about the estimate of effectiveness itself but about the quality of the underlying evidence that the number was derived from—are more difficult to assess and to communicate than a confidence interval [29].

The quality of the evidence base used to produce an estimate of effectiveness plays a crucial role in assessing how reliable and trustworthy the estimate is. If it is of high quality, the effectiveness information is likely more reliable and less subject to future change compared to when it is derived from low quality evidence. Systems, such as the GRADE working group evidence quality assessments [30–32] and the Effective Public Health Practice Project Quality Assessment Tool [33] have established ways both of assessing underlying quality of evidence

and attempting to produce concise ways to communicate such assessments, via descriptors such as 'low quality', 'moderate quality' or 'high quality' [34].

The limitations in the evidence around COVID-19 mitigation methods means that we are often faced with evidence that—when objectively assessed—is not high quality by any standard: usually rated between 'very low' and 'moderate' quality by GRADE. In fact, the same is often true of non-pharmaceutical interventions in other domains, such as physical exercise (e.g. [35]).

This produces a dilemma for public health communicators. Transparent and trustworthy evidence demands clear communication of both effectiveness estimates and the uncertainties around them, including cues of quality of evidence [36]. However, the evidence around the effects of communication of such uncertainty is mixed. Research shows that people prefer transparent communication about scientific uncertainty (i.e. direct/quantified uncertainty in this context) on COVID-19 [37]. In the context of communicating evidence from systematic reviews people appreciate being shown results quantitatively in a table with an indication of direct uncertainty [38, 39], and communicating quantified uncertainty around an effectiveness estimate (e.g. confidence intervals) often has only very small effects on the public's overall trust in either the estimate or the source of the message [40, 41]. However, little is known about the effects of communicating the quality of the underlying evidence behind estimates. Some work has assessed different presentation formats that include quality of evidence information, such as summary tables that follow GRADE guidelines on the display of quality of evidence information [38, 39] or comparisons of formats using letters versus symbols [42] and varying types of quality of evidence elements [43]; however, less empirical work has assessed how people react to the actual quality of evidence information they are given (i.e. the quality level), irrespective of the presentation format. To address this gap, in this research we asked whether communicating the quality of the underlying evidence (in the form of concise labels such as 'low' and 'high' quality of evidence) affects the public's trust in the information, their perception of the efficacy of the behavioural intervention being described, and the likelihood of them acting on the conclusions based upon the evidence (e.g. choosing to adopt the intervention).

In June 2020, The Lancet produced an infographic to accompany a meta-analysis of three behavioural interventions to protect against COVID-19 infection or transmission [6]. The infographic used many principles of good evidence communication based on empirical evidence, such as a clear comparison of control and intervention groups with absolute risks in comparable format and an icon array graphic to illustrate the simple percentages [44–49] (although the array was of an untested dimension of 5x10 rather than 10x10 icons). However, the graphic also included statements about the 'certainty of evidence' for each intervention (different from the original GRADE wording which used 'quality of evidence' but in line with GRADE's most recent phrasing [31, 34, 50, 51]), ranging from 'low' for eye protection and face masks to 'moderate' for physical distancing. Unlike other elements of the infographic, the inclusion of cues of certainty or quality of evidence have not been evaluated empirically in this context. We therefore used this infographic as a real-world setting for tackling the research questions outlined above.

## Overall materials and methods

These experiments were pre-registered (Experiment 1: https://aspredicted.org/blind.php?x=n6pd26; Experiment 2: https://osf.io/ag9th) and given ethical oversight by the Psychology Research Ethics Committee of the University of Cambridge (PRE.2020.086). US American adult participants were recruited through the survey panel provider company Respondi that is certified by the International Organization for Standardization (ISO) (respondi.com), and

were directed to a questionnaire in Qualtrics. With anonymised responses, the research therefore falls under the exemption category of the Common Rule policy of Protection of Human Research Subjects (2018) (45 Cfr 46) in the US meaning further permits and approvals in the country were not required. The recruitment platform is a web-based panel provider which notifies pre-registered individuals of upcoming studies for which they are eligible and compensates them for their time. We collected national quota samples matched to the US population on age and gender. Potential participants were asked their age and gender, and a quota system operated to allow only participants who fell into quotas not yet full to continue to the study. All participants gave written informed consent prior to participating in the research and were awarded a $1 equivalent in panel points.

Participants were randomised into experimental conditions via the Qualtrics randomisation function. Participants were randomised evenly to each experimental condition and were blinded to the study condition that they were randomized to.

Once participants entered the experimental groups, they were shown a version of an infographic and asked a series of questions about it. The versions were adapted from the original infographic described above, which illustrated the evidence for three potential methods of mitigating the transmission of the coronavirus: eye protection, face masks and physical distancing (https://www.eurekalert.org/news-releases/584789). We chose to study people's reactions to the presentation of information around eye protection because physical distancing and face masks were both already subject to much public and political discussion in the U.S. at the time of data collection (Sep 24–29, 2020 for Experiment 1; Oct 14–16, 2020 for Experiment 2) [52–55], so we anticipated that the audience may have prior beliefs around both of these measures which may affect their reactions to the experiment. The infographic was experimentally manipulated in Adobe Illustrator to produce different versions for testing.

Our key dependent variables were perceived trustworthiness of the information presented about the effectiveness of eye protection (index of three items, $\alpha_{Exp1} = 0.97$, $\alpha_{Exp2} = 0.96$, based on O'Neill's dimensions of trustworthiness, including competence and reliability [56]), perceived effectiveness of wearing eye protection, and likelihood of behavioural uptake, i.e. intentions to wear eye protection when in busy public places (each measured on a 7-point Likert scale). See Table 1 for the wording of the dependent measures. We collected further measures for exploratory purposes; the analysis of which are reported in the S1 File.

The questionnaire for both studies contained an attention check item: "Do you feel that you paid attention, avoided distractions, and took the survey seriously so far?" (answer options: *No, I was distracted; No, I had trouble paying attention; No, I didn't not take the study seriously*

**Table 1. Overview of dependent measures for Experiments 1 and 2.**

| Concept | Questions | Answer options |
|---|---|---|
| Perceived trustworthiness of information | How trustworthy do you think the information you saw on the effectiveness of eye protection is? | 7 point Likert scale, not trustworthy/accurate/reliable at all—very trustworthy/accurate/reliable |
| | How accurate do you think the information you saw on the effectiveness of eye protection is? | |
| | How reliable do you think the information you saw on the effectiveness of eye protection is? | |
| Perceived effectiveness of eye protection | How effective do you think eye protection is for reducing the chance of infection or transmission of COVID-19? | 7 point Likert scale, not effective at all—very effective |
| Behavioural uptake intentions | How likely are you to wear eye protection when in busy public places? | 7 point Likert scale, not at all likely—very likely |

*so far; No, something else affected my participation negatively; Yes*). Participants who failed the attention check, i.e. who gave an answer other than 'yes', were excluded as pre-registered. The attention check measure was administered prior to randomising participants into experimental treatment groups. Please refer to the S1 File for further details on materials and methods.

All analyses were carried out in R version 3.6 and the analysis code is available in the OSF repository.

## Experiment 1: Additional materials and methods

This experiment set out to test whether members of the general public reacted differently to different stated levels of quality/certainty of evidence (high versus low) and to the quality of evidence levels being described as 'quality of evidence' versus 'certainty of evidence', two alternate wordings used by GRADE. GRADE initially used the term 'quality of evidence'. More recently 'certainty of evidence' has become the preferred term [57] and is the one used in the original Lancet infographic. The experiment employed a 2x2 factorial design ('low' versus 'high' level of evidence x 'certainty' versus 'quality' wording) (Fig 1). We note that during production of the infographics, the right-most, grey column of the icon array for the chance of infection or transmission 'with eye protection' got cropped, leaving an array of 45 rather than 50 icons. We display the infographics as they were shown to participants. The icon array column was missing consistently for all experimental conditions. See the limitations section of the General Discussion for further discussion.

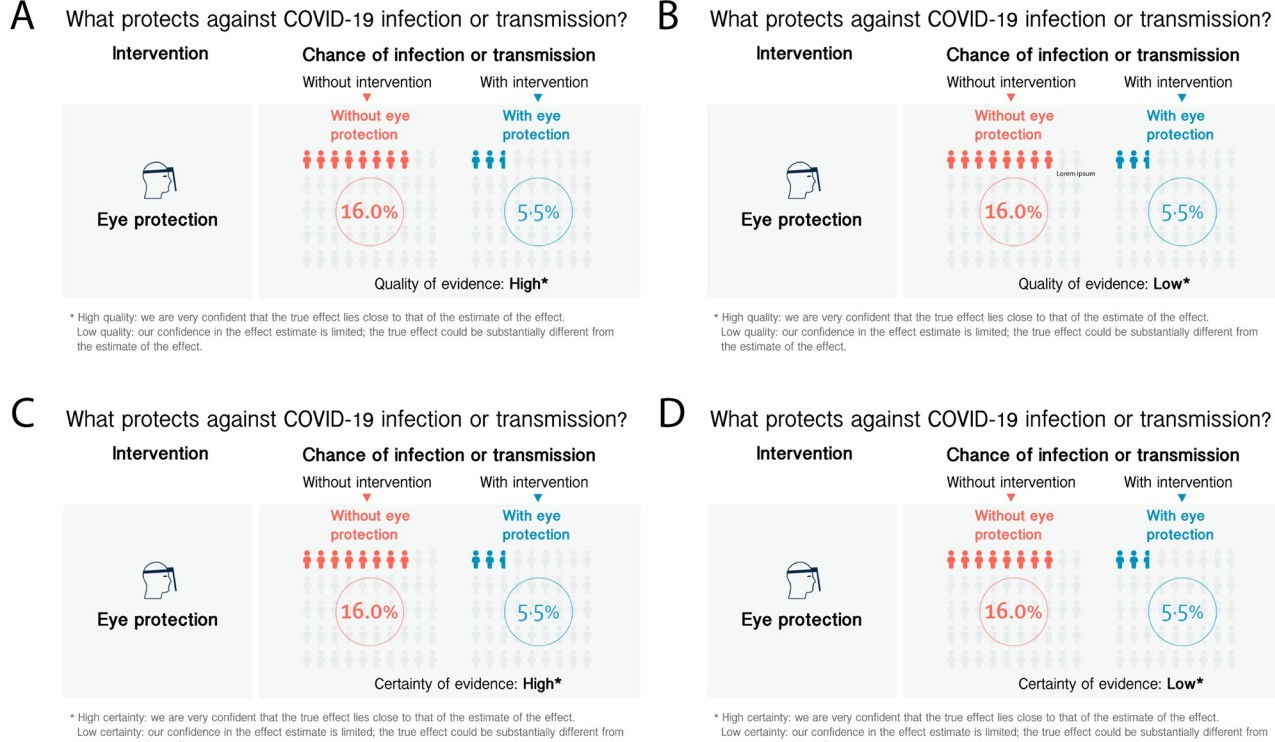

**Fig 1. Infographics used in Experiment 1 showing the chance of infection or transmission of COVID-19 without and with eye protection along with information on the underlying certainty/quality of evidence levels.** The four panels are depicting the infographics used for the four experimental conditions. (A) infographic shown to participants in the High Quality of evidence condition, (B) infographic shown to participants in the Low Quality of Evidence condition, (C) infographic shown to participants in the High Certainty of Evidence condition, (D) infographic shown to participants in the Low Certainty of Evidence condition.

Participants were shown the infographic once, and then asked a series of questions about it on different pages.

We hypothesized (see pre-registration) that people's trust in the information, their perception of the effectiveness of the intervention, and their likelihood of behavioural uptake would be higher for the group that is shown 'high' quality of evidence compared to the group that is shown 'low' quality of evidence information.

We pre-registered a sample of 949 participants, providing 95% power at alpha level 0.05 for small effects (f = 0.12). This target sample size included a buffer to account for attrition due to failing of the attention check. For data collection, we implemented real-time dynamic sampling which ensured that only those participants who passed the attention check were counted towards the analytic sample quotas. Therefore, the final number of participants for our analytic sample was the full pre-registered sample size.

## Experiment 1: Results

We sampled 949 participants (48.58% male, 51.42% female, $M_{age}$ = 45.25, $SD_{age}$ = 16.58; see further demographic details as well as number of participants in each experimental condition in Table 2). As pre-registered, we tested for main effects of quality of evidence level and wording for our various outcome measures.

Two-way Analysis of Variance (ANOVA) revealed a main effect of quality level ('high' versus 'low') on all three outcome measures, i.e. perceived trustworthiness of the information, perceived effectiveness of eye protection, and intentions to wear eye protection (Table 3). Due to the non-normal distribution of the data for the behavioural uptake outcome variable, regular ANOVA was complemented by non-parametric aligned ranks transformation ANOVA for both Experiments 1 and 2, which supported the parametric results for both studies (see Tables 3 and 7). As hypothesized, post-hoc testing using Tukey's honestly significant difference (HSD) test revealed that participants assigned to the 'low quality of evidence' infographic

**Table 2. Demographic details of participants in Experiment 1.**

| Variable | Overall (N = 949) | Low—Quality group (N = 243) | High—Quality group (N = 240) | Low—Certainty group (N = 223) | High—Certainty group (N = 243) |
|---|---|---|---|---|---|
| Gender, % | | | | | |
| Females | 51.42 | 49.79 | 50.83 | 52.47 | 52.67 |
| Males | 48.58 | 50.21 | 49.17 | 47.53 | 47.33 |
| Age, *Mean (SD)* | 45.25 (16.58) | 45.86 (16.03) | 44.74 (16.66) | 45.85 (17.27) | 44.59 (16.47) |
| Education, % | | | | | |
| Did not complete high school | 1.69 | 2.47 | 2.08 | 1.79 | 0.41 |
| High school degree or equivalent | 35.83 | 34.57 | 34.17 | 39.46 | 35.39 |
| Associate's degree | 14.44 | 16.87 | 10.42 | 13.90 | 16.46 |
| Bachelor's degree | 32.03 | 29.63 | 37.08 | 31.84 | 29.63 |
| Graduate or Professional degree | 16.02 | 16.46 | 16.25 | 13.00 | 18.11 |
| Political views, *Mean (SD)*[*] | 4.05 (1.67) | 4.06 (1.64) | 4.17 (1.71) | 4.00 (1.71) | 3.96 (1.64) |
| Numeracy, *Mean (SD)*[+] | 4.58 (1.78) | 4.56 (1.77) | 4.58 (1.71) | 4.56 (1.73) | 4.60 (1.91) |

[*]Political views on 7-point scale on spectrum from very left wing (or liberal) (1) to very right wing (or conservative) (7).

[+] Numeracy was measured using the sum of the scores of a combination of items (see supplementary information file for more details). The final 8-point scale ranges from low numeracy (1) to high numeracy (8).

**Table 3. Analysis of variance results for all outcome measures (Experiment 1).**

| | Quality level | Quality wording |
|---|---|---|
| Perceived trustworthiness | $F(1,946) = 35.61$, $p < .001$, $\eta_P^2 = 0.036$ | $F(1,946) = 0.01$, $p = .915$ |
| Perceived effectiveness | $F(1,946) = 17.4$, $p < .001$, $\eta_P^2 = 0.018$ | $F(1,946) = 0.40$, $p = .528$ |
| Behavioural intentions | $F(1,946) = 7.80$, $p = .005$, $\eta_P^2 = 0.008$ | $F(1,946) = 0.49$, $p = .484$ |
| Behavioural intentions—non-parametric aligned ranks transformation ANOVA | $F(1,945) = 7.12$, $p = .008$, $\eta_P2 = 0.007$ | $F(1,945) = 0.42$, $p = .518$ |

group indicated statistically significantly lower levels of perceived trustworthiness, perceived effectiveness, and intentions to wear eye protection compared to participants in the 'high quality of evidence' group (Table 4 and Fig 2).

No significant effect of wording ('certainty of evidence' versus 'quality of evidence') was observed across the three outcome measures (Table 3).

## Secondary analysis: Understanding

We pre-registered a secondary analysis to explore whether the difference in wording ('quality' versus 'certainty') affected people's understanding of the infographic. Understanding was measured via an index item of reported ease and completeness of comprehension of the effectiveness information in the infographic, as well as self-reported effort invested in understanding the effectiveness information (both measured on a 7 point Likert scale; see S1 File for details). An independent samples t-test revealed a small effect indicating that 'quality of evidence' was statistically significantly easier to understand for people compared to 'certainty of evidence'. Since the distribution of the measure was skewed, the parametric analysis was complemented by non-parametric testing for robustness purposes. Mann-Whitney test results were in line with the parametric findings (Table 5). Although descriptively people on average reported lower invested effort for the 'quality' wording compared to the 'certainty' wording, this difference was not statistically significant as indicated by an independent samples t-test with Mann-Whitney non-parametric follow-up (Table 5).

## Mediation analysis

We had hypothesized that communicating the quality of evidence level would influence people's perceived trustworthiness of the presented information, which could in turn affect people's intentions whether to wear eye protection. Specifically, we predicted that providing people with low (versus high) quality of evidence information would decrease people's trust and hence lead to lower intentions to wear eye protection. To formally test this hypothesis, we

**Table 4. Post hoc results for main effects of quality of evidence level and wording (Experiment 1).**

| | Low QoE | | High QoE | | | | p | Cohen's d |
|---|---|---|---|---|---|---|---|---|
| | Mean | 95% CI | Mean | 95% CI | Mean difference | 95% CI | | |
| Perceived trustworthiness | 4.11 | [3.96,4.26] | 4.74 | [4.60,4.88] | 0.62 | [0.42,0.83] | < .001 | 0.39 |
| Perceived effectiveness | 4.16 | [4.00,4.33] | 4.64 | [4.49,4.80] | 0.48 | [0.26,0.71] | < .001 | 0.27 |
| Behavioural intentions | 3.47 | [3.27,3.66] | 3.85 | [3.66,4.04] | 0.36 | [0.12,0.66] | 0.005 | 0.18 |
| | Quality of evidence | | Certainty of evidence | | | | | |
| | Mean | 95% CI | Mean | 95% CI | Mean difference | 95% CI | | |
| Perceived trustworthiness | 4.43 | [4.28,4.57] | 4.43 | [4.28,4.58] | 0.002 | [-0.21,0.21] | | |
| Perceived effectiveness | 4.37 | [4.21,4.53] | 4.45 | [4.29,4.61] | -0.08 | [-0.31,0.14 | | |
| Behavioural intentions | 3.61 | [3.42,3.80] | 3.72 | [3.52,3.91] | -0.13 | [-0.38,0.17] | | |

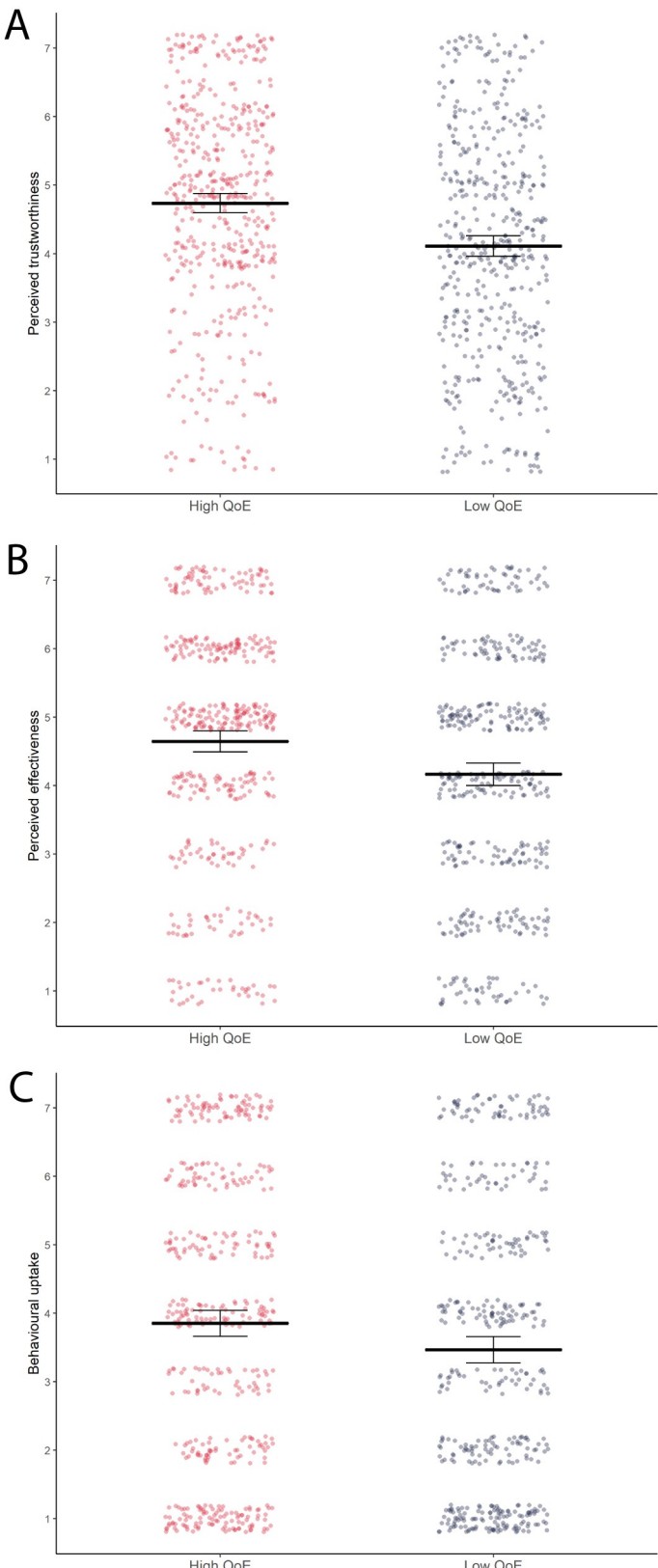

**Fig 2. The effects of giving a cue as to the certainty/quality of evidence (QoE) behind an effectiveness estimate on people's perceptions of the trustworthiness of the information (A), the perceived effectiveness of the intervention**

**(B), and the likelihood of them adopting it (C).** All dependent variables were measured on 7 point Likert scales ranging from 1-low to 7-high (please see Methods section for exact wording details for all measures). The three plots show mean effects and associated 95% confidence intervals (black horizontal lines and error bars), as well as underlying observed data distributions (coloured dotted points; red/left column = observations for high QoE groups, blue/right column = observations for low QoE groups). Data for all plots is depicted collapsed across wording conditions (quality/certainty).

pre-registered a mediation analysis of perceived trustworthiness on behavioural uptake intentions.

Mediation analysis was conducted using the mediation package in R [58], with parameter estimates based on 5000 bootstrapped samples for all reported results. Results support our hypothesis: There was a statistically significant direct effect of experimental condition on uptake intentions (b = -0.39, CI [-0.66, -0.12], p = .003) which was no longer statistically significant once the mediator was accounted for (b = 0.09, CI [-0.13, 0.31], p = .418). Importantly, the indirect effect of condition, i.e. low quality of evidence compared to high quality of evidence, on behaviour via perceived trustworthiness was statistically significant (b = -0.48, CI [-0.64, -0.32], p < .001).

### Additional secondary analyses

We also ran a range of other secondary and exploratory analyses as detailed in the pre-registration. These include the role of reported priors of effectiveness and quality of evidence perception for eye protection, (in-)congruency between priors and presented information, self-reported shifts in trust and behavioural intentions due to the infographic, effects of additional exploratory outcome variables, exploratory interaction analysis between experimental groups, and potential moderators of the observed experimental effects. The results of these additional exploratory analyses can be found in the S1 File. As one of these results, we found an interaction between numeracy and quality of evidence level on perceived trustworthiness and effectiveness, indicating that the effect of quality of evidence information on these outcomes depends to an extent on people's numeracy level. For participants with higher numeracy levels, we saw more pronounced differences in their responses to low versus high quality of evidence levels. This could hint at different levels of engagement with, or understanding of, the cues in the infographic between higher numeracy and lower numeracy participants. This partially motivated the design of Experiment 2.

### Experiment 1: Discussion

Experiment 1 suggested no difference between participants' reactions to the words 'quality' and 'certainty' when used to describe the underlying evidence base behind the use of eye protection in protecting against COVID-19 infection with regards to effects of perceived trustworthiness, perceived effectiveness or behavioural intentions. This is despite the fact that the

**Table 5. Quality versus certainty of evidence wording on understanding (Experiment 1).**

| | Quality of evidence | | Certainty of evidence | | Mean difference | 95% CI | | p | Cohen's d |
|---|---|---|---|---|---|---|---|---|---|
| | Mean | 95% CI | Mean | 95% CI | | | | | |
| Ease and completeness of comprehension | 5.68 | [5.56,5.80] | 5.47 | [5.34,5.60] | 0.21 | [0.03,0.38] | t(939.1) = 2.26 | 0.024 | 0.15 |
| Mann-Whitney test | | | | | | | W = 121759 | 0.026 | |
| Effort invested in understanding | 3.79 | [3.62,3.97] | 3.95 | [3.78,4.13] | -0.14 | [-0.40,0.09] | t(946) = -1.26 | 0.209 | |
| Mann-Whitney test | | | | | | | W = 107393 | 0.218 | |

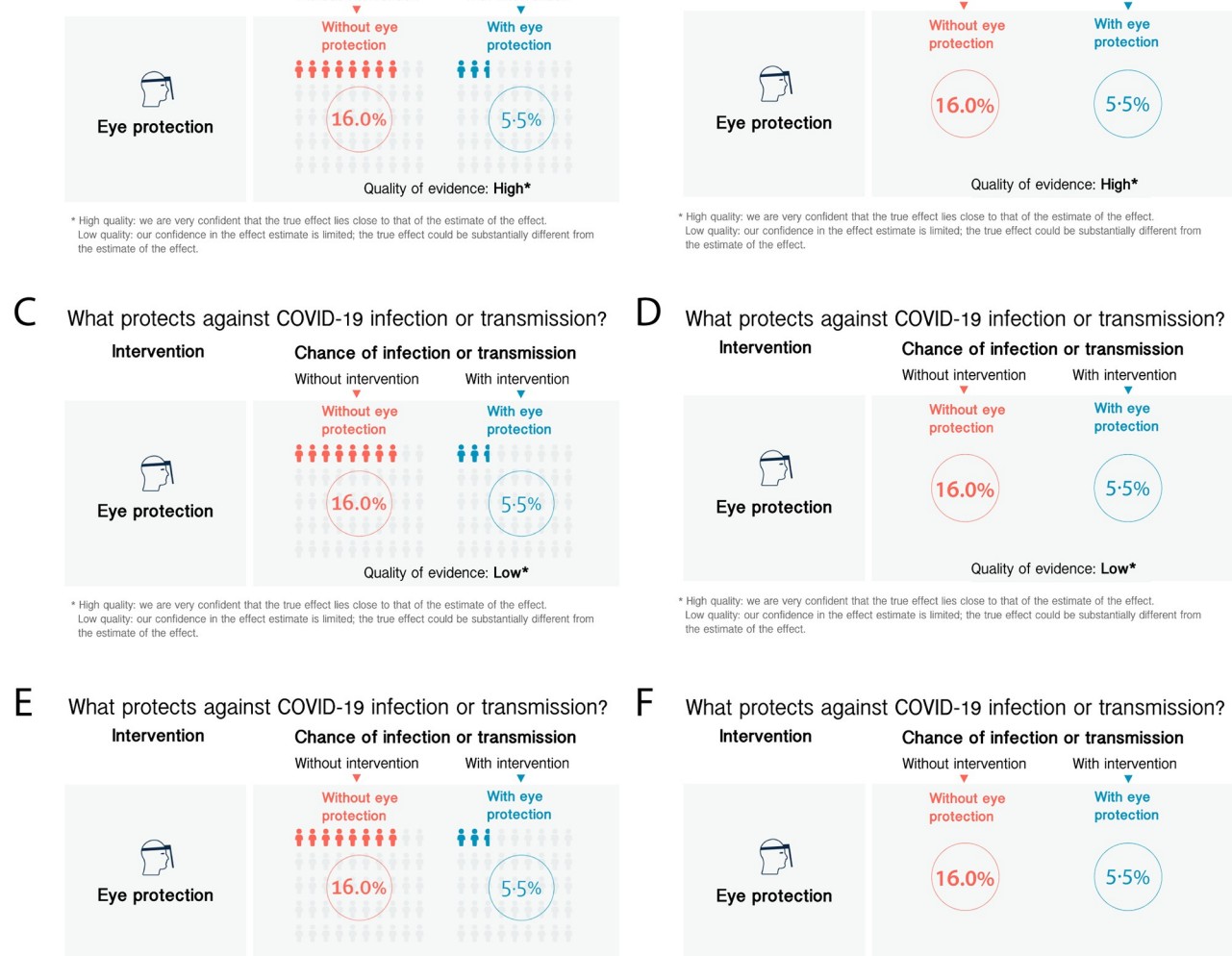

**Fig 3. Infographics used in Experiment 2 showing the chance of infection or transmission of COVID-19 without and with eye protection (in two information presentation formats) along with information on the underlying quality of evidence levels.** The six panels depict the infographics used for the six experimental conditions. (A) infographic shown to participants in the 'With Icon Array High Quality of Evidence' condition, (B) 'Without Icon Array High Quality of Evidence' condition, (C) 'With Icon Array Low Quality of Evidence' condition, (D) 'Without Icon Array Low Quality of Evidence' condition, (E) 'With Icon Array No Quality of Evidence' condition, (F) 'Without Icon Array No Quality of Evidence' condition.

phrase 'quality' could be seen as a more judgemental term (e.g. 'low quality' is more pejorative).

However, important differences arose in responses to the expressed quality or certainty level. A statement indicating high quality or certainty of evidence led to the information being trusted more than for a statement of low quality or certainty. Likewise, a statement of high quality or certainty led to people perceiving eye protection as more effective and indicating higher likelihood of wearing eye protection, than for a statement of low quality or certainty. Additionally, the difference in trust for the high compared to the low quality/certainty condition, appeared to influence the likelihood people said they would wear eye protection.

These results informed the design of Experiment 2, in which we left aside the 'certainty' wording and concentrated on investigating the effects of quality of evidence information, adding a condition in which participants were not given any cues as to the quality of the evidence.

The finding that higher numeracy participants may be more sensitive to differences in stated quality of evidence is interesting. Expert clinical guideline panels have been found to be more likely to make strong recommendations when the quality of the evidence is high [59], indicating that quality of evidence level plays a strong role in their decision making, and this may reflect a similar effect. We were keen to assess further the effects of comprehension on weighting of the information presented in the infographic (i.e. the estimated effectiveness of eye protection and the quality of the evidence the estimate was based on) given the differences in results seen between higher and lower numeracy participants. In reporting health information, it is very common to use numbers (such as percentages) for the effectiveness estimates but simpler, verbal cues for the quality of evidence information. It could be that differential levels of understanding of these cues affects their weighting in decision-making. We therefore wanted to attempt an experimental manipulation of the comprehensibility of the information to see whether this would in turn affect its perceived trustworthiness or the effect of the quality of evidence rating on perceived efficacy. In the infographic we tested, lower numeracy participants' understanding of the numbers might have been supported by graphics in the form of icon arrays (although one that visualized the percentages as number of people out of a total of 50 people instead of 100, hence going counter the presumably most intuitive visualization of a percentage). We therefore planned a condition in which this icon array was removed, to see whether this affected comprehension of the efficacy information and, in turn, people's assessment of that information.

## Experiment 2: Additional materials and methods

This experiment had two aims. Firstly, to test how participants assessed evidence quality when there was no statement regarding it, compared to evidence with an overt 'high' or 'low' quality label. Secondly, to assess whether participants' reactions to the two quality cues was altered when the icon array was removed, making the efficacy information purely numeric, potentially affecting participants' understanding.

We therefore used a 3 x 2 factorial design: three conditions of evidence quality cue ('high' versus 'low' versus no statement) and two conditions of information presentation formats (with and without icon array). See Fig 3. The infographics depicted are in the format in which participants saw them. Please refer to the materials and methods section of Experiment 1 for an elaboration on the graphical error that occurred in the production of the infographics, as well as the limitations section of the General Discussion for further discussion.

After randomisation, by contrast with Experiment 1, participants were shown the infographic above the questions on each page in this experiment to ensure that all participants had the information in front of them when indicating their responses, reducing potential of a memory recall or ability bias. Key dependent measures were the same as in Experiment 1 (see Table 1).

As for Experiment 1, we hypothesized that people's trust in the information, their perception of the effectiveness of the intervention, and their likelihood of behavioural uptake would be higher for the group that is shown the infographic with 'high quality of evidence' compared to the group that is shown the infographic with 'low quality of evidence'. We cautiously hypothesized, based on our experience of ongoing experiments in a different context, that the effects of the 'no quality of evidence' control group infographic would be closer to the 'high

 

**Table 6. Demographic details of participants in Experiment 2.**

| Variable | Overall (N = 1,191) | High QoE—with icon array group (N = 202) | Low QoE—with icon array group (N = 198) | No QoE—with icon array group (N = 200) | High QoE—without icon array group (N = 196) | Low QoE—without icon array group (N = 204) | No QoE—without icon array group (N = 191) |
|---|---|---|---|---|---|---|---|
| Gender, % | | | | | | | |
| Females | 51.47 | 50.00 | 55.05 | 55.50 | 52.04 | 52.45 | 43.46 |
| Males | 48.53 | 50.00 | 44.95 | 44.50 | 47.96 | 47.55 | 56.54 |
| Age, *Mean* (SD) | 45.31 (16.43) | 45.74 (16.32) | 46.12 (15.98) | 43.55 (15.52) | 44.36 (16.26) | 46.96 (17.02) | 45.06 (17.44) |
| Education, % | | | | | | | |
| Did not complete high school | 2.18 | 2.48 | 2.02 | 1.00 | 1.53 | 1.96 | 4.19 |
| High school degree or equivalent | 38.20 | 39.11 | 47.47 | 30.00 | 37.24 | 38.73 | 36.65 |
| Associate's degree | 15.28 | 13.37 | 16.16 | 18.00 | 14.29 | 14.22 | 15.71 |
| Bachelor's degree | 29.39 | 29.70 | 21.21 | 34.50 | 33.67 | 27.45 | 29.84 |
| Graduate or Professional degree | 14.95 | 15.35 | 13.13 | 16.50 | 13.27 | 17.65 | 13.61 |
| Political views, *Mean* (SD)* | 3.99 (1.62) | 3.88 (1.62) | 4.11 (1.69) | 4.06 (1.63) | 3.87 (1.65) | 4.01 (1.56) | 3.98 (1.60) |
| Numeracy, *Mean* (SD)+ | 4.34 (1.76) | 4.35 (1.76) | 4.18 (1.74) | 4.36 (1.67) | 4.34 (1.82) | 4.43 (1.78) | 4.40 (1.80) |

*Political views on 7-point scale on spectrum from very left wing (or liberal) (1) to very right wing (or conservative) (7).

+ Numeracy was measured using the sum of the scores of a combination of items (see supplementary information file for more details). The final 8-point scale ranges from low numeracy (1) to high numeracy (8).

quality of evidence' group compared to the 'low quality of evidence' group (see pre-registration).

We sampled 1191 participants providing 95% power, at alpha level 0.05 for small effects (f = 0.13, based on the results of Experiment 1). We implemented the same real-time sampling procedure checking for attention check fails as described in Experiment 1. The final number of participants for our analytic sample was therefore the full pre-registered sample.

**Table 7. Analysis of variance results for all outcome measures (Experiment 2).**

| | Quality level | Presentation format |
|---|---|---|
| Perceived trustworthiness | $F_{(2,1187)} = 15.05, p < .001, \eta_p^2 = 0.025$ | $F_{(1,1187)} = 0.53, p = .465$ |
| Perceived effectiveness | $F_{(2,1187)} = 9.98, p < .001, \eta_p^2 = 0.017$ | $F_{(1,1187)} = 2.68, p = .102$ |
| Behavioural intentions | $F_{(2,1187)} = 1.28, p = .279$ | $F_{(1,1187)} = 0.46, p = .496$ |
| Behavioural intentions—non-parametric aligned ranks transformation ANOVA | $F_{(2,1185)} = 1.17, p = .311$ | $F_{(1,1185)} = 0.01, p = .905$ |

**Table 8. Post hoc results for main effects of quality of evidence level and presentation format (Experiment 2).**

| | Low QoE | | High QoE | | No QoE | | Contrast | | | p | Cohen's d |
|---|---|---|---|---|---|---|---|---|---|---|---|
| | Mean | 95% CI | Mean | 95% CI | Mean | 95% CI | | Mean difference | 95% CI | | |
| Perceived trustworthiness | 4.07 | [3.91,4.24] | 4.68 | [4.53,4.84] | 4.48 | [4.32,4.63] | low—no | 0.43 | [0.18, 0.63] | 0.001 | 0.25 |
| | | | | | | | low—high | 0.61 | [0.39, 0.84] | < .001 | 0.38 |
| | | | | | | | high—no | -0.19 | [-0.42, 0.01] | 0.170 | |
| Perceived effectiveness | 4.11 | [3.94, 4.29] | 4.65 | [4.49, 4.81] | 4.45 | [4.27, 4.62] | low—no | 0.35 | [0.09, 0.57] | 0.018 | 0.19 |
| | | | | | | | low—high | 0.53 | [0.30, 0.78] | < .001 | 0.32 |
| | | | | | | | high—no | -0.20 | [-0.44, 0.03] | 0.203 | |
| Behavioural intentions | 3.48 | [3.27,3.69] | 3.72 | [3.50,3.93] | 3.55 | [3.34,3.76] | low—no | 0.10 | [-0.23, 0.36] | | |
| | | | | | | | low—high | 0.23 | [-0.06, 0.53] | | |
| | | | | | | | high—no | -0.15 | [-0.46, 0.13] | | |
| | With icon array | | Without icon array | | | | | | | | |
| | Mean | 95% CI | Mean | 95% CI | | | | Mean difference | 95% CI | | |
| Perceived trustworthiness | 4.45 | [4.32,4.58] | 4.37 | [4.24,4.50] | | | | 0.07 | [-0.11, 0.26] | | |
| Perceived effectiveness | 4.49 | [4.35,4.62] | 4.32 | [4.18,4.46] | | | | 0.16 | [-0.03, 0.36] | | |
| Behavioural intentions | 3.63 | [3.46,3.79] | 3.54 | [3.37,3.71] | | | | 0.07 | [-0.16, 0.33] | | |

## Experiment 2: Results

We analysed the results from 1191participants (48.53% male, 51.47% female, $M_{age}$ = 45.31, $SD_{age}$ = 16.43; see further demographic details in Table 6). As in Experiment 1, we pre-registered to test for main effects of quality of evidence level and format for our various outcome measures.

Two-way analysis of variance using Tukey HSD revealed a main effect of quality of evidence level on perceived trustworthiness of the information and perceived effectiveness of eye protection (Table 7), such that participants in the 'low quality of evidence' infographic group indicated statistically significantly lower levels of perceived trustworthiness and effectiveness compared to participants in the group that did not present quality of evidence information at all, as well as compared to those in the 'high quality of evidence' group. Participants in the 'high quality of evidence' infographic group did not statistically significantly differ in their trust or effectiveness perception from those in the group that did not receive quality of evidence information (Table 8 and Fig 4).

No statistically significant effects of quality of evidence level emerged for the behavioural uptake measure (Table 7).

No main effect of presentation format (with and without icon array) emerged, for any of the three outcome measures (Table 7).

### Understanding

We had hoped to explore the potential influence of people's understanding of the information given in the infographic (through having the two presentation formats represent different difficulty levels), especially its role in shaping the effects of the various levels of quality of evidence information on trust, perceived effectiveness and behaviour. However, a check to see whether our experimental manipulation of the infographic (removal of the icon array) had made a statistically significant difference to participants' self-reported understanding of the information (index item of reported ease and completeness of comprehension of the effectiveness information in the infographic) revealed that it had not ($t$(1188.9) = 0.04, p = .970; Wilcoxon rank sum test, W = 177244, p = .992).

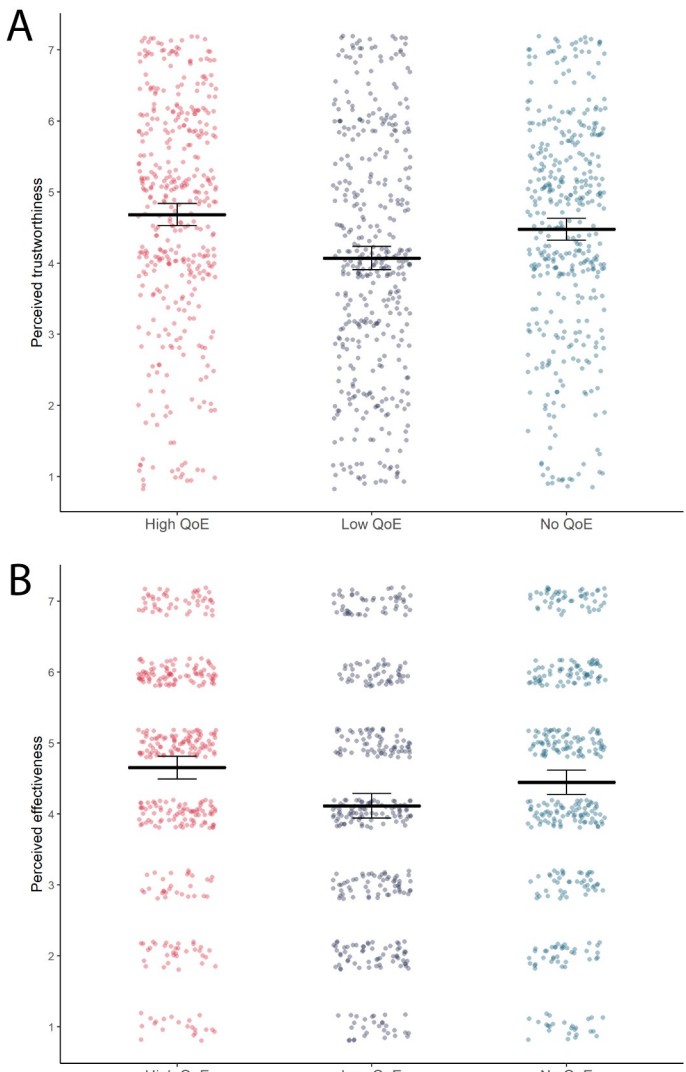

**Fig 4. The effects of giving a cue as to the quality of evidence (QoE) behind an effectiveness estimate on people's perceptions of the trustworthiness of the information (A) and the perceived effectiveness of the intervention (B).** All dependent variables were measured on 7 point Likert scales ranging from 1-low to 7-high (please see Methods section for exact wording details for all measures). The two plots show mean effects and associated 95% confidence intervals (black horizontal lines and error bars), as well as underlying observed data distributions (coloured dotted points; red/left column = observations for high QoE groups, blue/middle column = observations for low QoE groups, green/right column = observations for no QoE groups). Data for both plots is depicted collapsed across formatting conditions (with/without icon arrays).

For completeness, we still ran our pre-registered interaction analyses between quality of evidence level and format, as well as self-reported understanding. We investigated people's self-reported ease and completeness of understanding of the effectiveness information in the infographic, their objective understanding of the numeric effectiveness information, as well as the amount of effort they reported to have invested in understanding the information on the effectiveness of eye protection in the infographic. We did not find any statistically significant interactions for any of these potential moderators on any of our outcome measures. Detailed results and those of further exploratory analyses are reported in the S1 File.

## Experiment 2: Discussion

Experiment 2 replicated the main effects on two of our three dependent variables: there was a statistically significant effect of giving quality of evidence information on both the perceived trustworthiness of the information and the perceived effectiveness of the intervention. No statistically significant differences emerged for the behavioural uptake measure.

Experiment 2 furthermore extended the findings from Experiment 1: as hypothesized, we found that the effects of not giving people indications of the quality level of the evidence were similar to those seen in the 'high quality of evidence' infographic group, and statistically significantly different from those of the 'low quality of evidence' group.

This suggests that in the absence of explicit cues of the quality of the evidence, people responded to the information they were provided with as if it was high quality, and that only stating overtly that evidence is 'low quality' could significantly change people's perceptions. This might be because people implicitly assume a relatively high level of quality of evidence they are presented with in this kind of scenario (e.g. clearly presented estimates with no confidence intervals or cues of uncertainty in the evidence). Alternatively, it might be because people's implicit assumption is a 'neutral' level of evidence, which can be manipulated either up or down with an explicit cue, but that the cue of 'low quality' is much more salient, and people react to it more strongly compared to the 'high quality' cue, which does not make a statistically significant difference. The latter could be an example of loss aversion. Psychological theory has shown that people are more sensitive to losses compared to gains, which may cause low quality of evidence cues to pull effects downwards for the 'low' group more than high quality cues push it up [60–62].

When looking at the effects of degree of understanding on weighting of cues, unfortunately our manipulation of the format did not make a statistically significant difference to the understandability of the numerical effectiveness information, which could imply that the icon arrays were not making a positive difference in supporting the comprehension of the numbers presented, potentially due to shortcomings in the icon array (e.g. that the denominator used was 50 rather than 100 people, hence showing the 16% 'without eye protection' as 8 coloured-in icons and the 5.5% 'with eye protection' as 2.75 coloured-in icons), or that our measures of understandability were not sensitive enough to any differences.

## General discussion

Across two large, randomised experiments we show that information about the quality of underlying evidence changes public perceptions of estimates of the effectiveness of public health measures.

In Experiment 1 we show that a statement of high quality or certainty of evidence led to the information being trusted more than for a statement of low quality or certainty. In the same way it also affected how effective people judged eye protection to be in reducing the chance of COVID-19 infection, and the likelihood to which people indicated they would wear eye protection. Moreover, we show that effects on trust mediate the relationship between quality of evidence information and downstream behaviour (providing people with a statement of low quality of evidence decreased people's trust and in turn lowered their intentions to wear eye protection compared to a statement of high quality of evidence).

Looking at different phrasing (Experiment 1), we find no difference between participants' reactions to the words 'quality' and 'certainty' on measures of trust, perceived effectiveness or behavioural uptake intentions, although the two words are qualitatively different. It may be that people pay more attention to, or weight more, the qualifier (e.g. 'low' or 'high') than the terminology of the measure. We did find a small effect on understanding, such that

participants rated the term 'quality of evidence' to be easier to understand compared to 'certainty of evidence'. These empirical findings suggest that communicators might want to use the term 'quality'. We note however that (a) the effect size was small, (b) that we were not testing what participants actually understood by the two terms and so further research is warranted before conclusions can be drawn over which word is more appropriate.

Understanding people's reactions to public health communications gives important insights on factors affecting adoption and ultimately the success of non-pharmaceutical interventions. Although several studies have assessed the effects of non-pharmaceutical interventions in the context of COVID-19 in various countries [11–16], these studies have largely relied on modelling approaches using observational data, such as information on lockdown measures and other imposed restrictions and measures of COVID-19 prevalence (e.g. reproduction rates) [11, 12, 14, 16]. As we show through experimental randomised controlled trials, people's reactions to public health communication critically depend on their perceptions about whether they are being presented with high or low quality information, and this affects how much they trust the information, believe in the efficacy of the shown intervention, and, to an extent how likely they say they are to take action based on the communication.

By contrast with our findings here about the effects of 'indirect' uncertainty communication (as defined in [29] as uncertainty about the evidence underlying numerical estimates), experiments on the communication of 'direct' uncertainty (as defined in [29] as uncertainty about the actual numerical estimate itself), appears to have much less effect on trust, and full disclosure is preferred by the public [37, 40].

To our knowledge, this is the first published evidence on the effects of communicating ratings of the quality of evidence around health-related findings and as a result, an important ethical issue emerges from our findings. These experiments suggest that, in the absence of statement to the contrary, people treat information as if it is based on high quality evidence, and this affects their reactions to it. If estimates being communicated to the public are actually based on low quality evidence, lack of disclosure of this has implications: it could be seen to be misleading. The same could be considered true for non-numerical information, communicated as 'facts' or 'advice'.

In the case of individual medical decisions, where information is being given purely as a matter of informed consent or shared decision-making, the ethical (and sometimes legal) implications are clear: disclosure of the quality of the underlying evidence base is vital. However, in the realm of public health, where the mandate may be more to persuade than inform, it may be tempting for communicators to not disclose the low quality of evidence levels in connection with a recommendation or advice in order to promote 'compliance'. However, that is a decision that has to be made in the knowledge that that lack of disclosure is likely to affect people's reactions to the information and may be seen as unethical and infringing on autonomy. It has been argued that disclosure of uncertainties and honest communication of limitations to knowledge are vital for retaining public trust in the long run and for ensuring ethical medical science communication [63]. Recommendations and public health advice can be entirely justified even when there is a low underlying quality of the evidence (e.g. when there are also low risks to performing the action); however, in such cases it could be argued that the uncertainties should still be acknowledged, and the advice justified in a clear way. Such an approach may buffer any negative effects of the disclosure of low quality of evidence. We encourage further research to test such an approach empirically. In addition to the effects on a public audience, not acknowledging low quality of underlying evidence could inhibit further research to improve the evidence base.

This study is limited in that it tested only an online population in the US (albeit quota sampled), and only one health intervention. Further research could broaden this population and

context, for example, by using true probability samples, collecting data in multiple countries, engaging in field work, and testing a range of public health interventions. A further limitation is that the quality of evidence information provided in this research was a simple indication of the level, without further details as to the exact reasons for the rating. It would be useful to examine the effects of providing greater nuance and detail on the quality rating, in addition to the effects of adding an explanation for recommendations despite low quality of evidence as outlined above. Furthermore, our work only tested the provision of quality of evidence information in a text format. It is conceivable that providing a quality of evidence label in, for example, a graphical format akin to star ratings, might have a different effect. Lastly, our studies were designed to test overall effects across a broad population. Understanding potential differential effects on different subgroups of the population, such as low and high numeracy individuals, would help to complement knowledge on the effects of quality of evidence communication more broadly, and we thus encourage further research to investigate these relationships more deeply.

As mentioned in footnote 3, the images shown to participants showed an icon array that had had some of its icons cropped erroneously. This error was consistent across all studies and conditions and hence unlikely to introduce systematic bias that would affect our results in study 1. In study 2, where we tested in addition a difference in presentation format (with and without icon array display), a bias could have been introduced if participants noticed the varying amounts of light grey icons in the two icon array displays and were confused by it. We hence coded the free text responses that participants provided in both studies to identify any comments about the icon arrays. For neither study were there comments relating to confusion about the icon arrays. It therefore seems likely that participants did not notice the error, and we do not expect any influence of it on our observed effects.

## Supporting information

**S1 File.**
(DOCX)

## Author Contributions

**Conceptualization:** Claudia R. Schneider, Alexandra L. J. Freeman, David Spiegelhalter, Sander van der Linden.

**Data curation:** Claudia R. Schneider.

**Formal analysis:** Claudia R. Schneider.

**Investigation:** Claudia R. Schneider.

**Methodology:** Claudia R. Schneider, Alexandra L. J. Freeman, David Spiegelhalter, Sander van der Linden.

**Project administration:** Claudia R. Schneider, Alexandra L. J. Freeman.

**Validation:** Claudia R. Schneider, Alexandra L. J. Freeman, Sander van der Linden.

**Visualization:** Claudia R. Schneider.

**Writing – original draft:** Claudia R. Schneider, Alexandra L. J. Freeman.

**Writing – review & editing:** Claudia R. Schneider, Alexandra L. J. Freeman, David Spiegelhalter, Sander van der Linden.

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
