## [Decision Letter · Decision Letter 0]

16 Jun 2021

PONE-D-21-13365

The effects of quality of evidence communication on perception of public health information about COVID-19: two randomised controlled trials

PLOS ONE

Dear Dr. Schneider,

Thank you for submitting your manuscript to PLOS ONE. After careful consideration, we feel that it has merit but does not fully meet PLOS ONE’s publication criteria as it currently stands. Therefore, we invite you to submit a revised version of the manuscript that addresses the points raised during the review process.

I have been fortunate enough to receive two excellent reviews of your work. I would be grateful if you could submit a revised version of the manuscript addressing all the points that they make.

We look forward to receiving your revised manuscript.

Kind regards,

Richard Rowe

Academic Editor

PLOS ONE

Journal Requirements:

2. During our internal checks, the in-house editorial staff noted that you conducted research or obtained samples in another country. Please check the relevant national regulations and laws applying to foreign researchers and state whether you obtained the required permits and approvals. Please address this in your ethics statement in both the manuscript and submission information.

"The Winton Centre for Risk & Evidence Communication, thanks to the David & Claudia Harding Foundation"

5. Please amend your list of authors on the manuscript to ensure that each author is linked to an affiliation. Authors’ affiliations should reflect the institution where the work was done (if authors moved subsequently, you can also list the new affiliation stating “current affiliation:….” as necessary).

Reviewers' comments:

Reviewer's Responses to Questions

**Comments to the Author**

1. Is the manuscript technically sound, and do the data support the conclusions?

Reviewer #1: Yes

Reviewer #2: Yes

2. Has the statistical analysis been performed appropriately and rigorously? 

Reviewer #1: Yes

Reviewer #2: I Don't Know

3. Have the authors made all data underlying the findings in their manuscript fully available?

Reviewer #1: Yes

Reviewer #2: Yes

4. Is the manuscript presented in an intelligible fashion and written in standard English?

Reviewer #1: No

Reviewer #2: Yes

5. Review Comments to the Author

Reviewer #1: Very important study and well done. Congratulations to the authors. I enjoyed reading this work and think it provides useful information. However, I worry that the presentation of the data is its main limitation. Apologies for any typos or grammatical atrocities below.

Major Comments

1. Would you consider reporting per CONSORT and adhering to reporting standards for both trials please? http://www.consort-statement.org/consort-2010. This includes adding CONSORT flow diagrams.

2. Very long report and somewhat unusual reporting in that the format seems to be: 1. introduction, 2. general methods linked to 3 different online locations to understand the data (protocols, data, supplement), 3. Additional methods for experiment 1, 4. Experiment 1 results, 5. Experiment 1 discussion, 6. Additional methods for experiment 2, 7. Experiment 2 results, 8. Experiment 2 discussion. 9. Conclusions (another 2.5 pages!). Even for single reports of complex two trials, they much more brief. This paper should be reduced by at least 50% in length. There should be only one section for introduction, one for methods, one for results, and one for discussion.

2a. Description of outcomes - somewhat repetitive to have almost the exact same verbiage repeated for multiple headings (perceived trustworthiness, perceived effectiveness, etc.) when it could be summarized likely as a single line referring to multiple outcomes. eg. Compared to individuals randomized to an infographic displaying "high quality evidence", those assigned to the infographic displaying "high certainty evidence" reported x ("Quality" group mean [95%CI], "Certainty" group mean [95% CI]; mean difference mean [95%CI])...

3. Reporting of effects and interpretability - helpful to understand magnitude of effect such as mean difference with 95%CI rather than focusing on p values and technical description of statistical tests in abstract and main text. Very difficult for the general reader to interpret as is currently. No generalist will understand what "F(1,946)= 35.61" or "ηG2 = 0.036" or the like means. Likewise, reporting capital M (eg. "M = 4.11, 95% CI [3.96,4.26]" and similar uses of M, or "dadj", would not be clear or approachable by the general audience. Also not completely clear what OR represents (eg. line 226 or 234). If in fact an odds ratio, how exactly was this generated from the data and what does it represent? Presumably you would have had to make some kind of cut point? More helpful if you use plain language. Doesn’t mean you have to completely eliminate all the statistical reporting, but at minimum the text should be easily interpretable to a general reader.

Minor Comments

4. References mismatched? - lines 85-88, references 11-13 don't seem to match evidence quality rating systems. Please verify that all references match.

5. OSF database and protocols, authors - Spiegelhalter seems absent from the OSF database. I only see the other 3 coauthors. "Contributors: Claudia R. Schneider Alexandra Freeman Dr. Sander van der Linden". I recognize different indivdiuals contribute in different ways to projects. Please verify the correct contributions in OSF please.

6. line 93 - change "by the GRADE" to "by GRADE" or "using the GRADE approach" or something similar.

7. oxymoron - line 101-102 - "in general on COVID-19". Perhaps rephrase to "addressing interventions for COVID-19" or "addressing interventions for SARS-CoV2" or something similar.

8. Similar terms with two different meanings - "uncertainty" in the manuscript commonly refers to both (1) statistical variance associated with point estimates (eg. 95% CIs) and (2) quality (aka certainty or confidence) of estimates. Given the focus of uncertainty to refer to the latter rather than the former, is there verbiage different from "uncertainty" that can be consistently used throughout when referring to 95% CIs?

9. Abstract - please, preference is for estimates of effect rather than just p values presented.

10. Questions posed - For the concept of perceived trustworthiness, are "trustworthy" "accurate" and "reliable" synonymous? they may be correlated but do they truly reprsent the same construct? Trustworthiness would be the veracity that the data presented are true, eg. "a trustworthy source of information". One can generate a trustworthy and low certainty estimate of effect. One can also generate a untrustworthy, high certainty estimate of effect. By contrast, an estimate may be not accurate (the extent to which the reported value approximates a true value) nor reliable (the extent to which repeated measures of the same value result in the same estimate). I suppose clarity in exactly what "Trustworthy" was meant to convey and did you test at all that the people surveyed agreed with the intended meaning.

11. Any clustering or correlation of participants? eg. multiple members of same household? How did you (or the survey company you employed) handle this? Could you describe more how the survey company samples, precisely who the sampled population was, and how they are found by the company? Were individuals paid by the company to partake in the survey?

12. Line 183-184 - GRADE uses both certainty and quality of the evidence, and if anything, favours certainty of the evidence of quality of the evidence. All recent GRADE guidelines refer to "certainty". Hence, "the commonly used GRADE wording" is not be correct. Perhaps you could just say something to the effect of "two alternate wording used by GRADE".

13. Given that Experiment 1 was a 2x2 factorial, and Experiment 2 was a 3x2 factorial, please report whether there was any interaction first, then describe the effects of either assignment.

14. Supplemental tables describing demographic variables - Linked to the CONSORT main comment. Could this please be a main table encompassing both trials, as is often reported in randomized studies, and please report by assignment (+/- overall studied population) rather than solely overall studied population, as is presented currently.

15. Linked to reporting - summmary display of results - Do you think a simple table reporting the mean values (and SD or 95%CI) of each group per outcome, followed by their associated mean difference with 95%CI, might be more informative than Figures 2 and 4? (the dot plots could be moved to supplement)

16. eg.

17. A table with two column headings, Experiment 1 and Experiment 2 with their associated (brief) descriptions

18. Row headings are outcomes

19. Column 1-4 for Experiment 1 are mean & 95%CI values for groups 1-4 for the 2x2 trial. Column 5-9 are between group mean differences with 95%CI.

20. Similar could be repeated for Experiment 2.

21. "Tukey HSD" abbreviation is not spelled out.

22. Reporting of groups - Rather than describe groups as, for example, "in the high quality evidence group" it might be more clear to describe allocations as "participants assigned to the "high quality" infographic group" or "participants assigned to the infographic labelled as high quality", or similar.

23. Likewise, describing the effect of wording as quality or certainty of the evidence could be more clear. eg. "no main effect of quality wording" could be "no main effect of wording as quality or certainty of the evidence". Again more informative to describe mean differences between groups with 95%CI rather than emphasis on description of statistical tests and p value / hypothesis testing. Of course if you want to include the p value in additon to the estimate of effect, I would not oppose that.

24. Lines 245-259 - understanding of quality vs certainty of evidence - again, effect estimation may be more helpful here than just hypothesis testing. if the M values reported represent means, do you think a between group difference mean of 0.21 on a 7-point likert scale is meaningful? Could this be statistically signifincant but unimportant in practical terms?

25. Line 287 to 291 - The language is much too technical here and loses readers - myself included.

26. May be helpful to contrast these findings with those of non-public health settings. Do they apply to communication of scientific results to the public in general? How do clinicians or other stakeholders react to the same presentation formats (I understand you did not test this but is there any data to support these other scenarios)? With regards to clinical guideline panels, it is notable that they are more likely make strong recommendations when there is high certainty evidence (https://www.jclinepi.com/article/S0895-4356(21)00068-8/fulltext).

27. Would also be curious to know to what extent science communication should show mean effects with or without their associated 95%CI - a related topic to the one you tested. Any evidence in that regard?

Reviewer #2: This is a brilliant study. Congratulations.

I have some suggestions that you might want to consider.

Background

1. You may want to correct statements suggesting that the GRADE Working Group uses ‘quality of evidence’. Although it is correct that it did in the past, it has, for the most part, used ‘certainty of evidence’ since 2013 – including in the Lancet article that was the basis for the infographic you used. See, e.g., https://doi.org/10.1016/j.jclinepi.2012.05.011, https://doi.org/10.1016/j.jclinepi.2017.05.006, https://doi.org/10.1016/j.jclinepi.2018.01.013, https://doi.org/10.1016/j.jclinepi.2018.05.011, https://doi.org/10.1016/j.jclinepi.2021.03.026

2. You may want to be less obscure when you explain the concept of ‘certainty of evidence’ in the background (where you refer to ‘known unknowns’, deeper uncertainties, and ‘indirect’ uncertainties) and in the discussion (where, for reasons that are not obvious, you refer to imprecision of effect estimates as ‘direct’ uncertainty and other sources of uncertainty, without any explanation of what these are, as ‘indirect’). In the example you use in your study, GRADE was used to assess the certainty of the evidence, so you could quite easily refer to the sources of uncertainty considered in that approach (risk of bias, inconsistency, indirectness, imprecision, and reporting bias).

3. There is some other relevant research that you might want to reference in the background, e.g., https://doi.org/10.1016/j.jclinepi.2014.04.009, https://doi.org/10.1016/j.jclinepi.2007.03.011, https://doi.org/10.1177/0272989x10375853, https://doi.org/10.2196/15899

Methods

4. I would find it easier to understand your study if you described the methods in a more structured way. You may want to use the CONSORT checklist if you have not already.

5. You may want to provide more information about the participants, including your eligibility/selection criteria. I assume you only included adults. The Respondi website provides almost no information about their panel, so it is hard to know how typical or atypical they might be compared to other U.S. adults.

6. What were potential participants told about the study when they gave informed consent?

7. Why did you measure prior beliefs after showing participants the infographic? Did you check the reliability of their post hoc ‘prior’ belief by checking to see if change from those to their beliefs after seeing the infographic were consistent with their reported shift in beliefs?

8. The only description of the analysis that I found was that ‘All analyses were carried out in R version 3.6.’

9. Is there a reason why you did not make the protocol for these studies available together with the other material on OSF?

10. It is not clear why your hypothesis about no information about the certainty of the evidence being interpreted as high certainty evidence. That seems quite predictable given the way the infographic presented the information (with precise numbers appearing to give a clear answer to the question and no clue to suggest that the information was not trustworthy. Did you tell participants anything before they agreed to participate about what you were going to present that might have influenced how they perceived the information?

Also, I don’t follow your explanation for what you found in the discussion. There you say there were no ‘external’ cues about the certainty of the evidence, but there are cues that suggest it is trustworthy. I also don’t understand the difference between your first two explanations (implicitly assuming high certainty or assuming high certainty if low certainty is not explicit), and I don’t understand how loss aversion is relevant or would explain why participants perceived the infographic as representing high certainty evidence. (Moreover, the evidence suggests that framing effects may only occur under specific conditions - https://doi.org/10.1002/14651858.cd006777.pub2.)

Results

11. I found the statistics difficult to understand. It would be very helpful if you could make the results more understandable for non-statisticians – especially the size of the effects, including the meaning of the odd ratios you report.

12. There are several places where you use significant or significantly without being explicit that you are referring to statistical significance. That is problematic, since this commonly leads to misunderstanding. In addition, there are several places where you report no or not ‘significant effects’. When you do that, it is not possible to tell whether the findings are inconclusive because you did not have sufficient power to rule out a meaningful effect or they rule out a meaningful effect.

13. I would find a table helpful showing the main results for both experiments, i.e., the odds ratios (with confidence intervals) for each of the three main comparisons (low vs high certainty in experiment 1 and 2 and low vs no information about certainty in experiment 2) for trustworthiness, perceived effectiveness, and intended behaviour.

Discussion

14. The findings for trustworthiness and perceived effectiveness were very similar for experiment 1 and 2 (as one would expect). How do you explain the apparent difference in findings for intended behaviour?

15. Although you say that the infographic was “based on empirically-tested good practice recommendations for evidence communication”, I thought there were several problems with it, which you might want to consider in the discussion. First, it does not give any contextual information. The question you asked participants about behaviour was about their “intentions to wear eye protection when in busy public places”. However, all the studies on which the infographic are based were conducted in healthcare settings (except for one small study with no events). There is also no information about how long participants were followed up. The baseline risk (16%) is very high and misleading. It is highly unlikely that anyone not using eye protection in busy public places would experience that high a risk of getting Covid-19. It also is unlikely that random types of eye protection used by ordinary people in crowded public places would have that large of an effect. There is no information about potential harms or downsides. I found the icon array confusing rather than helpful, since I did not understand immediately that each icon represented two people. Also, I am unaware of evidence that indicates that icon arrays effectively communicate information about effects. There is at least one other study that found that an infographic (including an icon array) did not improve understanding (compared to a plain language summary): https://doi.org/10.1016/j.jclinepi.2017.12.003.

16. You may want to reconsider or better justify your recommendation that communicators use the word ‘quality’ instead of ‘certainty’ based on participants reporting that it was ‘easier to understand’. I do not agree with your recommendation. First, are you confident that they understood what the word means in this context correctly? I suspect more people are likely to misinterpret the meaning of ‘quality’, given how the word is commonly used, than they are to misinterpret ‘certainty’. Second, whatever term is used, it should always be accompanied by an explanation, since most people are unlikely to understand the basis for judgements about the certainty of the evidence. Third, there is evidence that suggests that people want information about the certainty of the evidence, that they want plain language text – not just numbers, and that using adjectives such as ‘probably’ and ‘may’ can help to communicate the certainty of the evidence (https://doi.org/10.1016/j.jclinepi.2014.04.009, https://doi.org/10.1177/0272989x10375853, https://doi.org/10.1016/j.jclinepi.2019.10.014). I suspect that is more likely to be important for communicators to do (consistently) than it is for them to use ‘quality’ instead of ‘certainty’.

17. You may want to strengthen your discussion of what you call an ethical dilemma. First, I don’t think public health authorities often do or should present information about effects in isolation. In fact, they frequently fail to provide any quantitative information. So, to the extent there is a dilemma, it is more in relation to including information about the certainty of evidence in connection with a recommendation or advice. In that context, there can be good reasons for recommending something, despite low certainty evidence, and that justification can and should be provided. That has been done by many organisations and people that have made strong recommendations while being transparent about the uncertainty of the evidence for pandemic control measures. Second, many if not most people would consider it unethical to, for example, withhold information about adverse effects of vaccines to persuade people to be vaccinated. How is it different to knowingly withhold information about the certainty of the evidence to persuade people? If you believe there are good reasons to persuade people to wear eye protection in crowded public places, then you should make an honest argument to persuade them. How is it ethically justified to be dishonest and knowingly present untrustworthy information as though it were trustworthy? Third, when there is important uncertainty or disagreement, not being honest and transparent can perpetuate practices that are wasteful and may be harmful. It also can inhibit research to reduce uncertainty and disagreement, and it can undermine trust. Persuasive tactics can infringe on people’s autonomy if information is withheld or if persuasive tactics are not justified. They can also inadvertently harm people. If there is a good justification to persuade people, the basis for doing so should be transparent and persuasive messages should not distort the evidence. This does not mean that clear, actionable messages cannot stand alone. Key messages should be up front, using language that is appropriate for targeted audiences, but it should be easy for those who are interested to dig deeper and find more detailed information, including the justification for a recommendation. When there are important uncertainties, they should be acknowledged.

Best,

Andy Oxman

6. PLOS authors have the option to publish the peer review history of their article (what does this mean?). If published, this will include your full peer review and any attached files.

Reviewer #1: No

Reviewer #2: No

---

## [Author Response · Author response to Decision Letter 0]

6 Aug 2021

PONE-D-21-13365

The effects of quality of evidence communication on perception of public health information about COVID-19: two randomised controlled trials

PLOS ONE

Response to Reviewers

Journal Requirements:

Thank you for sharing the style templates. The manuscript has been updated to meet PLOS ONE’s style requirements.

2. During our internal checks, the in-house editorial staff noted that you conducted research or obtained samples in another country. Please check the relevant national regulations and laws applying to foreign researchers and state whether you obtained the required permits and approvals. Please address this in your ethics statement in both the manuscript and submission information.

The Psychology Research Ethics Committee of the University of Cambridge which provided ethical oversight for the studies reported in this manuscript was aware of the fact that the samples would consist of US American participants. Likewise, all participants were made aware in the informed consent that they would be participating in research conducted at the University of Cambridge, UK. The research falls into the exemption category of the US’s 2018 Common Rule policy for Protection of Human Research Subjects (45 Cfr 46) as it is “Research involving benign behavioral interventions in conjunction with the collection of information from an adult subject through verbal or written responses” with individual participant anonymity. This meant that further approvals and permits were not required.

We have added a sentence to the ethics statement to clarify this.

"The Winton Centre for Risk & Evidence Communication, thanks to the David & Claudia Harding Foundation"

Thank you for making us aware of this. The funding information has been removed from the manuscript. Thank you for updating our Funding Statement in the online submission form for us. The amended statement should read:

“This project was carried out using core funding from The Winton Centre for Risk & Evidence Communication, which itself is funded by a donation from the David & Claudia Harding Foundation.”

Unfortunately, I was unable to link my ORCID iD during the initial submission process. I just followed the steps you provided but received the following error message:

“An identical, Validated ORCID already exists in the database. An ORCID may be linked to only one record in the database, and so you may have registered another user record in the past and retrieved/validated your ORCID for use with that.”

The only option I have is to click ‘Ok’. No ORCID is added to my profile.

If you are able to add my ORCID iD from your end to the system, or to merge the account to which my ORCID iD is already attached to the one that I am using for this submission that would be fantastic. My ID is:

https://orcid.org/0000-0002-6612-5186

Thank you!

5. Please amend your list of authors on the manuscript to ensure that each author is linked to an affiliation. Authors’ affiliations should reflect the institution where the work was done (if authors moved subsequently, you can also list the new affiliation stating “current affiliation:….” as necessary).

All authors on the manuscript are linked to an affiliation, two to two affiliations. I don’t see any missing affiliations in the manuscript file. I don’t assume that the problem is that authors cannot have two affiliations; in our case they are held concurrently.

Each author also has an ORCiD – if there is a way of adding these, that would also be great.

A section ‘Supporting information’ with a caption for the supporting information file has been added to the end the manuscript as detailed in the style guide. In-text citations of the supporting information have been updated throughout the manuscript accordingly.

 

Reviewers' comments:

Reviewer's Responses to Questions

Comments to the Author

1. Is the manuscript technically sound, and do the data support the conclusions?

Reviewer #1: Yes

Reviewer #2: Yes

2. Has the statistical analysis been performed appropriately and rigorously?

Reviewer #1: Yes

Reviewer #2: I Don't Know

3. Have the authors made all data underlying the findings in their manuscript fully available?

Reviewer #1: Yes

Reviewer #2: Yes

4. Is the manuscript presented in an intelligible fashion and written in standard English?

Reviewer #1: No

Reviewer #2: Yes

 

5. Review Comments to the Author

Reviewer #1: Very important study and well done. Congratulations to the authors. I enjoyed reading this work and think it provides useful information. However, I worry that the presentation of the data is its main limitation. Apologies for any typos or grammatical atrocities below.

Thank you for the positive and helpful feedback.

Major Comments

1. Would you consider reporting per CONSORT and adhering to reporting standards for both trials please? http://www.consort-statement.org/consort-2010. This includes adding CONSORT flow diagrams.

Thank you for raising this point. Our work is ‘standing between disciplines’, in that it is social sciences/psychology research (as opposed to clinical trials) that at the same time likely has an interested readership spanning both the social sciences and the medical/public health field. 

We understand that the disciplines of medicine/public health and psychology/social sciences have developed very different styles of reporting. 

In writing the manuscript, we followed the reporting guidelines of the American Psychological Association (APA), one of the most widely used guidelines in the social sciences, where multiple experiments need to be reported separately each with their own section. 

We tried to bridge the gap between our social sciences research and reporting in a way that is also clear for a more medically/public health oriented readership by using the CONSORT 2010 checklist (as linked on the CONSORT website (http://www.consort-statement.org/consort-2010)) for the reporting of the studies to ensure that all applicable checklist items are included either in our manuscript or the supplementary information file. The online submission system did not ask to upload the completed checklist. We are happy to do so if useful, if PLOS ONE could provide an upload link or we can send it via email; whichever is most convenient.

We appreciated the suggestion of adding the flow diagrams to provide an easy, visual overview of the experiments’ conditions and participant numbers. Thank you for that. Given that our studies are not typical clinical trials, we drew on comparable social sciences research publications that included CONSORT flow diagrams (e.g. Woloshin and Schwartz, 2011) to craft diagrams for our work. We added two CONSORT flow diagrams, one for each of the reported studies, to the supporting information.

Woloshin, S., & Schwartz, L. M. (2011). Communicating data about the benefits and harms of treatment: a randomized trial. Annals of internal medicine, 155(2), 87-96.

2. Very long report and somewhat unusual reporting in that the format seems to be: 1. introduction, 2. general methods linked to 3 different online locations to understand the data (protocols, data, supplement), 3. Additional methods for experiment 1, 4. Experiment 1 results, 5. Experiment 1 discussion, 6. Additional methods for experiment 2, 7. Experiment 2 results, 8. Experiment 2 discussion. 9. Conclusions (another 2.5 pages!). Even for single reports of complex two trials, they much more brief. This paper should be reduced by at least 50% in length. There should be only one section for introduction, one for methods, one for results, and one for discussion.

We appreciate this valuable input and suggestions for making the paper more concise and accessible. 

We renamed the conclusion section ‘general discussion’ – which is really what it is. Thank you for making us aware of it. 

In addition to following APA guidelines where multiple experiments need to be reported separately each with their own section, when we tried reporting all the methods together, and all the results together, the flow and readability of the manuscript was compromised. Since Experiment 2 was created based on the findings and insights from Experiment 1, it is helpful for readers to understand the conclusions we drew from Experiment 1, to then see why Experiment 2 was necessary. We used the ‘general methods’ section to avoid repetition of methodological information that was common to both.

However, we agree that the writing could be shortened. We attempted to do that. In particular, we omitted all sections that were not absolutely relevant to understanding the methods of the studies and consolidated separate results reporting for the three outcome measures into one paragraph for both experiments, in addition to omitting most statistics in the text, instead referring to results tables. These measures shortened the manuscript significantly. We believe it is now more easily accessible with a better flow. We hope you agree. If not, we’d be happy to implement any suggestions you have for further improvement. Thank you.

2a. Description of outcomes - somewhat repetitive to have almost the exact same verbiage repeated for multiple headings (perceived trustworthiness, perceived effectiveness, etc.) when it could be summarized likely as a single line referring to multiple outcomes. eg. Compared to individuals randomized to an infographic displaying "high quality evidence", those assigned to the infographic displaying "high certainty evidence" reported x ("Quality" group mean [95%CI], "Certainty" group mean [95% CI]; mean difference mean [95%CI])...

Thank you, this is a good point and you are absolutely right that the results reporting was a bit repetitive. Thank you for bringing this to our attention. As noted above, we have condensed the results sections to report findings for all three outcome measures in one paragraph, i.e. omitting our subheadings. In order to preserve legibility and understandability we removed all main statistics reporting from the text and placed them into tables, to which we refer in the writing. We hope this will help with creating a less repetitive flow for the reader.

3. Reporting of effects and interpretability - helpful to understand magnitude of effect such as mean difference with 95%CI rather than focusing on p values and technical description of statistical tests in abstract and main text. Very difficult for the general reader to interpret as is currently. No generalist will understand what "F(1,946)= 35.61" or "ηG2 = 0.036" or the like means. Likewise, reporting capital M (eg. "M = 4.11, 95% CI [3.96,4.26]" and similar uses of M, or "dadj", would not be clear or approachable by the general audience. Also not completely clear what OR represents (eg. line 226 or 234). If in fact an odds ratio, how exactly was this generated from the data and what does it represent? Presumably you would have had to make some kind of cut point? More helpful if you use plain language. Doesn’t mean you have to completely eliminate all the statistical reporting, but at minimum the text should be easily interpretable to a general reader.

We entirely agree with you that the results should be easily interpretable. 

As outlined above, we had closely followed the reporting requirements and standards of the American Psychological Association (APA) for social sciences research. As our work is bridging between the social sciences and medical field, we appreciate that a ‘hybrid’ approach might be needed, and tried to implement it following your valuable suggestions.

We left the reporting of p-values and the reporting of the model statistics in the paper as we believe it is important for full and transparent reporting of data analysis, however, as described above, we moved most of the statistics into tables to help a general reader who is not interested in numbers to better focus on the flow of the prose explanations.

We also exchanged all ‘M’ abbreviations with the word ‘mean’ and changed the effect size reporting to Cohen’s d. We had initially reported an adjusted Cohen’s d using Hedge’s correction for small sample sizes (Hedge’s g). However, given that our sample sizes are >200 per experimental group and that Cohen’s d and Hedge’s g converge at larger N, we decided to change the reporting to Cohen’s d for ease of understanding and clarity.

We report means per group and 95% confidence intervals. In addition, we now report mean differences between groups with 95% confidence intervals as requested. We hope this will help to make our manuscript understandable and useful to a general audience.

The abbreviation OR does indeed stand for ‘odds ratio’. We converted Cohen’s d into an odds ratio*, as we thought it might be useful to have odds ratio as an additional effect size metric in the paper to support understanding. We did not base our analytical/inferential models on the odds ratio (we did not run logistic/likelihood models). We now realize that it created more confusion than assistance and have taken out all odds ratios from the paper.

*Conversions of d into OR were based on:

Borenstein, M., Hedges, L. V., Higgins, J. P. T., & Rothstein, H. R. (2009). Converting among effect sizes. Introduction to meta-analysis, 45-49.

Minor Comments

4. References mismatched? - lines 85-88, references 11-13 don't seem to match evidence quality rating systems. Please verify that all references match.

Thank you for catching this oversight. We have adjusted the references.

5. OSF database and protocols, authors - Spiegelhalter seems absent from the OSF database. I only see the other 3 coauthors. "Contributors: Claudia R. Schneider Alexandra Freeman Dr. Sander van der Linden". I recognize different indivdiuals contribute in different ways to projects. Please verify the correct contributions in OSF please.

Prof Spiegelhalter does not have a OSF account, but we have now added him as a named contributor on the project.

6. line 93 - change "by the GRADE" to "by GRADE" or "using the GRADE approach" or something similar.

Thank you for catching this. Done.

7. oxymoron - line 101-102 - "in general on COVID-19". Perhaps rephrase to "addressing interventions for COVID-19" or "addressing interventions for SARS-CoV2" or something similar.

Thank you. Done.

8. Similar terms with two different meanings - "uncertainty" in the manuscript commonly refers to both (1) statistical variance associated with point estimates (eg. 95% CIs) and (2) quality (aka certainty or confidence) of estimates. Given the focus of uncertainty to refer to the latter rather than the former, is there verbiage different from "uncertainty" that can be consistently used throughout when referring to 95% CIs?

Thank you for bringing this up. The different kinds of uncertainty are indeed critical to our work. In order to make the two types we are addressing here explicit, we attempt to explain the different ‘types’ of uncertainty in the second paragraph of the Introduction: quantified ‘direct’ uncertainty around an estimate and unquantified ‘indirect’ uncertainty around the evidence behind an estimate. This is meant to give the reader an understanding of the two concepts and how they relate.

“Attempts at quantification of their effectiveness (e.g. How much does wearing eye protection reduce the chance of infection or transmission of COVID-19?) leads to a number of levels of uncertainty. Any experimental or observational data can give a point estimate (e.g. a percentage point reduction in the chance of infection or transmission) with a confidence interval (‘direct’ uncertainty, as defined in [10]). Meta-analyses can combine such estimated ranges, but the quantified uncertainty in confidence intervals only reflects a certain amount of the total uncertainty at play. Systematic biases, such as stemming from shortcomings in study design or data collection processes, unexplored variation and a host of other factors that are not easily quantified and cannot be deduced from the effectiveness estimate and its confidence interval, cause more ‘indirect’ uncertainties. These ‘indirect’ uncertainties – not directly about the estimate of effectiveness itself but about the quality of the underlying evidence that the number was derived from – are more difficult to assess and to communicate than a confidence interval [10].”

In other instances where we use the word ‘uncertainty’ we include a clarification ‘follow up’ to explain, as for instance here where we say ‘including cues of quality of evidence’:

“Transparent and trustworthy evidence demands clear communication of both effectiveness estimates and the uncertainties around them, including cues of quality of evidence[17].”

And in the Discussion we now recap the difference between the two where we contrast the finding here in ‘indirect’ uncertainty with previous work on ‘direct’ uncertainty:

“By contrast with our findings here about the effects of ‘indirect’ uncertainty communication (as defined in [10] as uncertainty about the evidence underlying numerical estimates), experiments on the communication of ‘direct’ uncertainty (as defined in [10] as uncertainty about the actual numerical estimate itself), appears to have much less effect on trust, and full disclosure is preferred by the public [18, 20].”

We hope that our writing is now clear enough to distinguish the different types of uncertainty to the reader but if not, we are happy and grateful to receive specific indications where we could change our language to make things more clear. Thank you.

9. Abstract - please, preference is for estimates of effect rather than just p values presented.

Thank you. We added effect sizes to the abstract in addition to the p-values. We think both together provide valuable information. If you, however, prefer to remove all statistics from the abstract, we are happy to do so as well.

10. Questions posed - For the concept of perceived trustworthiness, are "trustworthy" "accurate" and "reliable" synonymous? they may be correlated but do they truly reprsent the same construct? Trustworthiness would be the veracity that the data presented are true, eg. "a trustworthy source of information". One can generate a trustworthy and low certainty estimate of effect. One can also generate a untrustworthy, high certainty estimate of effect. By contrast, an estimate may be not accurate (the extent to which the reported value approximates a true value) nor reliable (the extent to which repeated measures of the same value result in the same estimate). I suppose clarity in exactly what "Trustworthy" was meant to convey and did you test at all that the people surveyed agreed with the intended meaning.

Thank you for raising this critical point. Our multi item measure of trustworthiness is based on O’Neill’s (2018) dimensions of trustworthiness which include competence and reliability. We have added this note to the manuscript. These measures are not synonymous in reality (depending on the situation as you lay out), but in this case we are investigating participants’ psychological perception of the situation. In the absence of any further information about the accuracy or reliability of the estimates, the only information they have ‘to hand’ to make their judgments is the information they are given about the quality of evidence. This probably explains why in these experiments we find high item intercorrelations in our samples and Cronbach’s alphas of 0.96 and 0.97 (reported in the manuscript) showing high internal consistency, indicating that people judge these concepts very similarly in this context.

O’Neill, O. (2018). Linking Trust to Trustworthiness. International Journal of Philosophical Studies, 26(2), 293–300. https://doi.org/10.1080/09672559.2018.1454637

11. Any clustering or correlation of participants? eg. multiple members of same household? How did you (or the survey company you employed) handle this? Could you describe more how the survey company samples, precisely who the sampled population was, and how they are found by the company? Were individuals paid by the company to partake in the survey?

Respondi is a panel provider that is certified by the International Organization for Standardization (ISO). It uses both off-line and online recruitment strategies for its actively managed online access panel. Participants receive credit for participation in online surveys. Respondi draws a random sample from the panel, which is then stratified and matched on national quotas. Notifications to eligible participants are sent depending on the need for different quotas set for each demographic group. We employed a quota system which allowed us to match the participants to the US population on age and gender. We did collect quite detailed demographic data, which allows us to look at who the participants are, and our experience of Respondi is that they give a fairly good match in the US to socio demographic features, although of course, being an online platform, it does exclude some participant demographics where online access or literacy is very low. We have now added more details about the recruitment process and the platform to the manuscript.

12. Line 183-184 - GRADE uses both certainty and quality of the evidence, and if anything, favours certainty of the evidence of quality of the evidence. All recent GRADE guidelines refer to "certainty". Hence, "the commonly used GRADE wording" is not be correct. Perhaps you could just say something to the effect of "two alternate wording used by GRADE".

Thank you very much for bringing this to our attention. We updated this section following your advice.

13. Given that Experiment 1 was a 2x2 factorial, and Experiment 2 was a 3x2 factorial, please report whether there was any interaction first, then describe the effects of either assignment.

Thank you for raising this point. We did not anticipate/hypothesize an interaction between quality of evidence wording and level (Experiment 1) or between presentation format and quality of evidence level (Experiment 2). Hence, both experiments were pre-registered only to test main effects and not interactions for the main analysis. 

A pre-registration in the social sciences is done prior to data collection, so upon the design stage of the research. It outlines the study design, hypotheses, exclusion criteria, analyses and other aspects of the research. The pre-registration is then uploaded onto a pre-registration site (commonly used sites in the social sciences are OSF or AsPredicted), and time stamped. After that it cannot be changed. The purpose for it is to support replicable science and avoid that researchers change their analyses approaches/hypotheses after they have collected the data. It is thus important to follow the pre-registration for data analysis. Given that we pre-registered to test main effects for our main analysis, not interactions, we stuck to that to not violate the pre-registration. 

After all pre-registered analyses are reported, it is customary to then report any additional/secondary/exploratory analyses (such as, in this case, interactions) either in the manuscript or in the supporting information. For Experiment 1 we pre-registered to run exploratory interaction analyses between quality level and quality terminology. We report these exploratory analyses in the supporting information, in the section “Interaction analysis between quality of evidence level and terminology”.

14. Supplemental tables describing demographic variables - Linked to the CONSORT main comment. Could this please be a main table encompassing both trials, as is often reported in randomized studies, and please report by assignment (+/- overall studied population) rather than solely overall studied population, as is presented currently.

Thank you for this excellent suggestion. We agree that it would be helpful for the reader to get an overview of the demographics per experimental group assignment in addition to the overall sample. We have added a column for each experimental condition, to report the demographic variables for each group, and moved these tables into the main manuscript as suggested. However, we think it would be cumbersome to combine the demographics for each experiment into a single table (making it quite massive and possibly hard to decipher given the amount of experimental groups), so have kept them separate for the moment. If you disagree, we are happy to change this.

15. Linked to reporting - summmary display of results - Do you think a simple table reporting the mean values (and SD or 95%CI) of each group per outcome, followed by their associated mean difference with 95%CI, might be more informative than Figures 2 and 4? (the dot plots could be moved to supplement) eg A table with two column headings, Experiment 1 and Experiment 2 with their associated (brief) descriptions Row headings are outcomes, Column 1-4 for Experiment 1 are mean & 95%CI values for groups 1-4 for the 2x2 trial. Column 5-9 are between group mean differences with 95%CI. Similar could be repeated for Experiment 2.

Thank you very much for this brilliant suggestion. We agree that tables will help with making the prose text of the manuscript less ‘cluttered’ and more accessible. We’ve added tables providing the statistical results for both Experiments 1 and 2 to the manuscript. These tables also serve the purpose of allowing us to cut down on the results reporting in the text as commented on above. We now refer to these tables in the manuscript text when presenting our results. We would prefer to keep the graphical visualizations of the results in the main paper as we believe they add value and help with understanding (‘quick glance’ overview of the results) for a general audience readership. If you disagree, we are happy to move them to the supplementary information.

21. "Tukey HSD" abbreviation is not spelled out.

Thank you for catching this. It is now spelled out at first mention.

22. Reporting of groups - Rather than describe groups as, for example, "in the high quality evidence group" it might be more clear to describe allocations as "participants assigned to the "high quality" infographic group" or "participants assigned to the infographic labelled as high quality", or similar.

Thank you. We adjusted the language accordingly in the manuscript.

23. Likewise, describing the effect of wording as quality or certainty of the evidence could be more clear. eg. "no main effect of quality wording" could be "no main effect of wording as quality or certainty of the evidence". Again more informative to describe mean differences between groups with 95%CI rather than emphasis on description of statistical tests and p value / hypothesis testing. Of course if you want to include the p value in additon to the estimate of effect, I would not oppose that.

Thank you for these suggestions for improvement. We’ve adjusted the wording describing the main effect of wording as quality or certainty of evidence throughout the manuscript to be more clear. We’ve added mean differences and 95% CIs to the results reporting. We now report all statistics; means, mean differences, along with 95% CIs, as well as statistical test results in the results tables for both Experiments 1 and 2.

24. Lines 245-259 - understanding of quality vs certainty of evidence - again, effect estimation may be more helpful here than just hypothesis testing. if the M values reported represent means, do you think a between group difference mean of 0.21 on a 7-point likert scale is meaningful? Could this be statistically signifincant but unimportant in practical terms?

Thank you for raising these points. Yes, ‘M’ does stand for ‘mean’ (we had followed APA reporting standards which uses the abbreviation). We have adjusted this to write out ‘mean’ though in the results tables to be more clear. We present an estimation of effect size in addition to the significance testing results. For the effect of wording on understanding we report a Cohen’s d of 0.15, which is as a ‘rule of thumb’ commonly classified as a ‘small effect’. 

The meaning in terms of practical or ‘real world’ relevance of small effect sizes is a very good question and in fact one that pertains to many areas of research in social psychology and the social sciences more generally, as observed effect sizes are often small. However, even small effect sizes can be meaningful when considering them on a population level. For instance, it might be that there is no big difference in understanding between ‘quality’ and ‘certainty’. However, if a decision maker on a practical level were to make a choice on use of terminology, merely considering the aspect of understanding for the example here, then it would be useful to know that there is a difference, albeit small, in order to help guide decision making. We therefore believe that reporting even small significant effect sizes can be relevant and meaningful in practical terms, in addition to making our findings available for other researchers to replicate and investigate further. We agree, though, that such small effect sizes should not necessarily be the basis for firm guidelines.

25. Line 287 to 291 - The language is much too technical here and loses readers - myself included.

Thank you for making us aware of this. We agree that the language used should be accessible. We have updated the language in this section to explain the interaction results between quality of evidence level and numeracy more clearly (we hope).

26. May be helpful to contrast these findings with those of non-public health settings. Do they apply to communication of scientific results to the public in general? How do clinicians or other stakeholders react to the same presentation formats (I understand you did not test this but is there any data to support these other scenarios)? With regards to clinical guideline panels, it is notable that they are more likely make strong recommendations when there is high certainty evidence (https://www.jclinepi.com/article/S0895-4356(21)00068-8/fulltext).

This is an interesting point and something that would certainly be worthwhile to test, i.e. whether the findings we observed for quality of evidence effects in a health domain, would be the same in another domain. Unfortunately, our data cannot speak to domain generalizability versus differences.

The findings you pointed out, that clinical guidelines panels are more likely to make strong recommendations when there is high certainty of evidence may be aligned with what we observe for numeracy; that higher numeracy participants may be more sensitive to differences in stated quality of evidence. We included this citation in the discussion of Experiment 1. Thank you for making us aware of it.

27. Would also be curious to know to what extent science communication should show mean effects with or without their associated 95%CI - a related topic to the one you tested. Any evidence in that regard?

This is a very interesting question. As we write in the introduction there is some work that suggests that exposing people to direct uncertainty (e.g. through showing confidence intervals), seems to not negatively affect measures of trust for instance.

“communicating quantified uncertainty around an effectiveness estimate (e.g. confidence intervals) often has only very small effects on the public’s overall trust in either the estimate or the source of the message [19, 20].”

A possible explanation might be that people understand that science is inherently uncertain and that predictions can have ranges and that they therefore don’t see the showing of uncertainty intervals as a ‘bad’ or ‘non trustworthy’ thing. If anything it might show honesty and transparency with regards to the accuracy of data. Surely a very important area for further research.

We’d like to thank you for taking the time to go through our manuscript so diligently. The manuscript has improved substantially due to your excellent and valuable comments, we very much appreciate it.

 

Reviewer #2: This is a brilliant study. Congratulations.

I have some suggestions that you might want to consider.

Thank you for the positive feedback and for the very helpful and valuable comments.

Background

1. You may want to correct statements suggesting that the GRADE Working Group uses ‘quality of evidence’. Although it is correct that it did in the past, it has, for the most part, used ‘certainty of evidence’ since 2013 – including in the Lancet article that was the basis for the infographic you used. See, e.g., https://doi.org/10.1016/j.jclinepi.2012.05.011, https://doi.org/10.1016/j.jclinepi.2017.05.006, https://doi.org/10.1016/j.jclinepi.2018.01.013, https://doi.org/10.1016/j.jclinepi.2018.05.011, https://doi.org/10.1016/j.jclinepi.2021.03.026

Thank you very much for bringing this to our attention. We removed the wording ‘the commonly used GRADE wording’ and exchanged it for saying ‘two alternate wordings used by GRADE’ (in addition to providing a footnote which elaborates on the change in terminology). Thank you also for sharing these excellent references with us which we used to strengthen the writing.

2. You may want to be less obscure when you explain the concept of ‘certainty of evidence’ in the background (where you refer to ‘known unknowns’, deeper uncertainties, and ‘indirect’ uncertainties) and in the discussion (where, for reasons that are not obvious, you refer to imprecision of effect estimates as ‘direct’ uncertainty and other sources of uncertainty, without any explanation of what these are, as ‘indirect’). In the example you use in your study, GRADE was used to assess the certainty of the evidence, so you could quite easily refer to the sources of uncertainty considered in that approach (risk of bias, inconsistency, indirectness, imprecision, and reporting bias).

Thank you. We edited the introduction and discussion to make the two concepts, i.e. direct or quantified uncertainty versus indirect or unquantified uncertainty more clear. GRADE ratings themselves, we believe, include the ‘direct’ uncertainty (i.e. the confidence interval – the uncertainty around the estimates) as part of the overall GRADE rating, and so combine what we define as two different concepts (direct and indirect uncertainty) into one. This is a level of complexity that we hoped to avoid discussing in the manuscript as it didn’t seem germane to the issues we were exploring, but it does mean that we didn’t want to go into detail about what GRADE ratings actually contain in terms of causes of uncertainty.

3. There is some other relevant research that you might want to reference in the background, e.g., https://doi.org/10.1016/j.jclinepi.2014.04.009, https://doi.org/10.1016/j.jclinepi.2007.03.011, https://doi.org/10.1177/0272989x10375853, https://doi.org/10.2196/15899

Thank you very much for bringing this research to our attention. We incorporated these references in the introduction.

Methods

4. I would find it easier to understand your study if you described the methods in a more structured way. You may want to use the CONSORT checklist if you have not already.

Thank you for raising this point. We have used the CONSORT checklist to ensure that all applicable elements in the checklist are included in the manuscript and supplementary information. Some of the details might be reported in a slightly different order or in different sections from the checklist. For instance, we split the reporting into Experiment 1 (methods, results, discussion) then Experiment 2 (methods, results, discussion) to help with clarity and understandability of the paper and following reporting guidelines of the American Psychological Association (APA) where multiple experiments need to be reported separately each with their own section. We tried out putting all methods in one section and all results. However, this cluttered the paper and made it quite hard to follow the logic of the two sequential experiments as a reader. 

We understand that the disciplines of medicine/public health and psychology/social sciences have developed very different styles of reporting. 

We had initially followed APA guidelines as outlined above but have now attempted to bridge the gap towards medicine/public health, although the type of research doesn’t fall naturally into a ‘clinical trial’ format.

We have now further condensed some of the reporting and repetition in the results sections and pulled out the statistics into tables, to make the prose easier to follow. 

In addition, we have now included CONSORT flow diagrams into the supplementary information as suggested by another reviewer.

We hope that these steps will help with ease of understanding for readers from all discipline backgrounds.

5. You may want to provide more information about the participants, including your eligibility/selection criteria. I assume you only included adults. The Respondi website provides almost no information about their panel, so it is hard to know how typical or atypical they might be compared to other U.S. adults.

Thank you for catching this. Yes, we only recruited adults (i.e. individuals aged at least 18). We have added this to the participant description in the manuscript, along with some more details about the Respondi recruitment process.

Because we implemented a quota system, this allowed us to match the participants to the US population on age and gender. While of course not a fully representative sample, this provides a significant increase in sample quality compared to non-representative online samples. We have also now included tables in the main manuscript including more demographic details of participants in the experiments following the suggestion of another reviewer, allowing readers to assess representativeness.

6. What were potential participants told about the study when they gave informed consent?

Participants were invited to partake in an academic research study ‘on perceptions of COVID-19 risk and information about COVID-19’. They were told that the study was exploring how different people are perceiving the risk of COVID-19 and was aiming to learn more about what people are doing in response to the risk of COVID-19. Participants were also told that they would be asked to provide some information about themselves. Other parts of the information sheet included outlining of possible benefits/risks in taking part, information about confidentiality, a GDPR statement, approximate duration of the study and contact details in case of questions; as mandated by the Psychology Research Ethics Committee of the University of Cambridge. Participants were not told any details of the study design, e.g. that they would see information on quality of evidence or that they would be randomized to seeing different infographics, in order to not compromise the study design and not introduce confounds. In the debrief at the end of the study participants were again provided with the contact details of one of the team members to reach out to in case of any questions, in addition to having the opportunity to write any comments into a feedback box.

The consent section will be included into the survey measures documents on OSF to be available for the interested reader. Upon acceptance of this paper for publication (when all changes made during the review process are finalized), we will create a new section on the OSF repository site entitled ‘published paper materials – [month] 2021’. We will upload the adjusted analysis code there (with the additions made in the review process), a pdf print of the two pre-registrations (to have them all in one place for the user’s convenience), the data, as well as the survey measures including the consent section. We will move all materials that are currently on the OSF repository site into a section entitled ‘pre-registration materials – April 2021’ to clearly distinguish which materials reflect the updated ones from the peer review process.

7. Why did you measure prior beliefs after showing participants the infographic? Did you check the reliability of their post hoc ‘prior’ belief by checking to see if change from those to their beliefs after seeing the infographic were consistent with their reported shift in beliefs?

The ‘priors’ were collected at the end of the survey in order to not prime people ahead of the experimental manipulations, i.e. to not introduce confounds into the studies. This of course means that we cannot be sure that they are true ‘priors’ as they might be affected by the experimental manipulations (i.e. seeing the infographics and answering questions about it). We discuss this in the supporting information in the section ‘The role of priors – effectiveness and quality of evidence views’ for both Experiments 1 and 2, where we write: 

“[…] it is therefore possible that people’s answers were influenced by the experimental groups they had been assigned to. For instance we observe that people’s perceived effectiveness as an outcome measure of our experimental manipulation is highly and significantly correlated with their views on the effectiveness of eye protection supposedly prior of being shown our experimental information (r = 0.81, t(947) = 42.61, p < .001; Spearman’s rank correlation rho = 0.80, p < .001). We thus advise caution in the interpretation of the results of our ‘priors’ analysis, and encourage future research to measure priors ahead of time for a more reliable picture. We report the analyses here nevertheless, in order to satisfy the pre-registration.”

As we state, we do provide the results of these exploratory analyses for completeness.

For exploratory purposes we furthermore included two measures of self-reported match between people’s priors of the effectiveness of eye protection and the effectiveness level shown in the infographic, as well as between people’s priors of the underlying quality of evidence level and the quality of evidence level indicated in the infographic. We report these in the sections ‘Self-reported (in-)congruency between priors and presented info’ in the supporting information. In this section we explain that it seems that people’s responses were influenced by the experimental manipulation and that people might not be able to or might not want to report on their priors after having received information. 

We write:

“[…] It is thus difficult to tell from the measures as we had implemented them in this study whether their prior perceptions of the effectiveness levels of eye protection and underlying quality of evidence were in fact congruent with the presented information or not. We encourage further research that measures priors in advance of any experimental manipulation.”

We also included exploratory measures of people’s self-reported shift in trust and behavioural intentions due to the infographic; results of which are presented in the sections ‘Self-reported shift in trust and behavioural intentions’ in the supporting information. We describe that the distributions of both items show that most people indicated that the infographic did not have an influence on their trust and behaviour and discuss that this may suggest that to an extent people might not be aware of the influence that the infographic has on them, or that they might not want to admit any influence. Nevertheless, comparing group averages on reported trust and behavioural shifts due to the infographic reveals a significant difference between the high and low quality of evidence groups. We fully present these results in the supporting information sections for transparency although we do have a lot of caution in their interpretation.

8. The only description of the analysis that I found was that ‘All analyses were carried out in R version 3.6.’

In addition to detailing the statistical software used for data analysis, we describe all statistical models and tests used in the respective Results sections. E.g. “Two-way Analysis of Variance (ANOVA) revealed…”, “Since the distribution of the measure was skewed, the parametric analysis was complemented by non-parametric testing for robustness purposes. Mann-Whitney test results were in line with the parametric findings…”, “Mediation analysis was conducted using the mediation package in R [30], with parameter estimates based on 5000 bootstrapped samples for all reported results…”. It may be that our reporting is clearer in this revised version of the manuscript where we have used tables to report findings.

The analysis code is also freely available in the OSF repository.

If you have anything in particular in mind that you feel is missing from the analysis reporting, please let us know and we are happy to amend.

9. Is there a reason why you did not make the protocol for these studies available together with the other material on OSF?

For social science/psychology research the commonly used type of ‘protocol’ for research studies are ‘pre-registrations’. These include information on the research questions being asked or hypotheses being tested, the key dependent variables and how they are measured, how many and which conditions participants will be assigned to, which analyses will be conducted to examine the research questions/hypotheses, how outliers will be defined and handled, rules for excluding observations, how many observations will be collected and what will determine sample size, and info on secondary/exploratory analyses. Both our experiments are pre-registered and the links are detailed in the manuscript. The first study is pre-registered on AsPredicted (https://aspredicted.org/blind.php?x=n6pd26 ), the second study is pre-registered on OSF (https://osf.io/ag9th). We had to switch to OSF for the second pre-registration as there is a word limit for the AsPredicted site. Study materials, such as questionnaires and data sets, are commonly made available in repositories, such as OSF. Our data and materials are available on OSF (https://osf.io/z6ps9/ ). So OSF can function as a data repository and also site for pre-registrations.

For user convenience we will add a pdf print of the two pre-registrations to the main OSF project repository site, section ‘published paper materials – [month] 2021’ – as outlined above in the response to question 6.

10. It is not clear why your hypothesis about no information about the certainty of the evidence being interpreted as high certainty evidence. That seems quite predictable given the way the infographic presented the information (with precise numbers appearing to give a clear answer to the question and no clue to suggest that the information was not trustworthy. Did you tell participants anything before they agreed to participate about what you were going to present that might have influenced how they perceived the information?

Also, I don’t follow your explanation for what you found in the discussion. There you say there were no ‘external’ cues about the certainty of the evidence, but there are cues that suggest it is trustworthy. I also don’t understand the difference between your first two explanations (implicitly assuming high certainty or assuming high certainty if low certainty is not explicit), and I don’t understand how loss aversion is relevant or would explain why participants perceived the infographic as representing high certainty evidence. (Moreover, the evidence suggests that framing effects may only occur under specific conditions - https://doi.org/10.1002/14651858.cd006777.pub2.)

Participants were not given any information that pertained to the infographic or the experimental conditions they were assigned to prior to partaking in our surveys. 

Our hypothesis concerning the effect of the ‘no quality of evidence’ group was based on findings in other similar work that we conducted. We detailed this in the pre-registration as follows:

“We are furthermore exploring effects when participants are not presented with any quality of evidence information at all. In other similar work we have seen that people tend to assume rather high quality of evidence in the absence of quality of evidence information. These studies were set in a different context though (less immediate self-relevant behavioural context). We therefore cautiously hypothesize that effects of the ‘no quality of evidence’ control group are closer to the ‘high’ quality of evidence group compared to the ‘low’ quality of evidence group; but are aware that effects may be different due to the contextual caveat outlined.”

Our speculation as to the reason of a potential effect of the ‘no quality of evidence’ group being closer in level on the outcome measures to the ‘high quality of evidence’ group compared to the ‘low quality of evidence group’ was that in the absence of any quality of evidence information people might assume it’s of high quality. The potential explanations for this which you outline might certainly play a role here, i.e. that if the infographic presents information in a clear fashion, without including direct uncertainty ranges for instance, people might just not have any indicator that the information might not be based on high quality evidence. 

We had already indicated this to an extent in the manuscript by mentioning the absence of external cues to quality:

“This might be because people implicitly assume a relatively high level of quality of evidence they are presented with in this kind of scenario (i.e., in the absence of external cues of its quality)”

With external cues here we mean the absence of explicit quality of evidence information, i.e. the ‘no quality of evidence’ group. As you suggest, the absence of other cues that might indicate low quality (e.g. in the language that is used, the way the numbers are shown etc.) might not give people any reason to ‘doubt the quality’. These are good points and we added details from your suggestions to make this point more clear and complete. Thank you.

With regards to our interpretative explanation of the findings of the ‘high quality of evidence’ group being at similar levels as the ‘no quality of evidence group’, we suggested that either a) people may have ‘implicit goodwill’ in a sense, meaning they just assume high quality unless told otherwise, or b) people might be at a neutral level but then react more strongly to a negative cue than a positive cue – which would be predicted by Prospect theory and loss aversion – i.e. if they get told that something is explicitly of low quality (‘loss dimension’) they adjust their ‘neutral’ level sharply downward. For the ‘gains dimension’ however, people don’t have such strong reactions, and hence stating explicitly that it is ‘high quality’ does not drive the outcome measure levels upwards enough to lead to a statistical difference between the ‘high’ and ‘no quality of evidence’ groups. We have added some more language to the manuscript to explain this more.

These were interpretative explanations that we offered as part of the discussing of our results and putting them into context of relevant theory. We merely wanted to highlight that this could be a potential mechanism here. Of course any claims to be made beyond our interpretive suggestion of the role that prospect theory could play here would need further and targeted empirical testing. We have added a note to the manuscript to make this clear. 

Unfortunately, we were unable to review the reference you cited as the link is not working. We would be happy to review it if you could provide the authors/paper title. Thank you.

Results

11. I found the statistics difficult to understand. It would be very helpful if you could make the results more understandable for non-statisticians – especially the size of the effects, including the meaning of the odd ratios you report.

We have reformatted the Results sections to increase clarity and understandability. We now report all main statistics in tables and not in the manuscript text to make the flow of the writing easier to read. We initially reported two measures of effect size in the manuscript, i.e. Cohen’s d and a conversion of Cohen’s d to odds ratios*, intended to support understandability for a wider audience. However, as the other reviewer pointed out, we would have had to cut our continuous response scale in order to have just two responses for a calculation of odds ratios. Such a change of our data/design doesn’t seem to be an adequate route to pursue. To avoid any confusion, we have taken out all odds ratios from the paper (see also our response to the other reviewer’s comment above).

Additionally, we have changed our effect size reporting to report Cohen’s d throughout. We had initially reported an adjusted Cohen’s d using Hedge’s correction for small sample sizes (Hedge’s g). However, given that our sample sizes are >200 per experimental group and that Cohen’s d and Hedge’s g converge at larger N, we decided to change the reporting to Cohen’s d for ease of understanding and clarity.

*Conversions of d into OR were based on:

Borenstein, M., Hedges, L. V., Higgins, J. P. T., & Rothstein, H. R. (2009). Converting among effect sizes. Introduction to meta-analysis, 45-49.

12. There are several places where you use significant or significantly without being explicit that you are referring to statistical significance. That is problematic, since this commonly leads to misunderstanding. In addition, there are several places where you report no or not ‘significant effects’. When you do that, it is not possible to tell whether the findings are inconclusive because you did not have sufficient power to rule out a meaningful effect or they rule out a meaningful effect.

We entirely agree. This is an important distinction. We have exchanged all mentioning of ‘significant’ to ‘statistically significant’ in the manuscript text to make this clearer.

Both experiments were powered at 95%, at alpha level 0.05 for small effects (f=0.12 for Experiment 1, f=0.13 for Experiment 2). This means we were powered at 95% to detect effects larger than f=0.12/0.13, so the reported non-significant effects provide evidence against the existence of a meaningful (f=0.12/0.13-sized or larger) effect. 

We acknowledge of course that our estimates of the effect size are estimates. It is of course possible that the true effect size is smaller. 

However, under the assumption that our estimated effect size is true and given our sample size, we thus believe that our non-significant findings can be viewed as providing some level of disconfirming evidence. 

Additional testing can be carried out to provide more information on the possible null effect. Equivalence testing for instance can assess whether an effect is outside the range of the smallest effect size of interest as defined by the researcher. For instance, when performing equivalence testing using the effect size that we powered for in Experiment 2 (f=0.13 or d=0.26), to look into the statistically non-significant effect of presentation format on perceived trustworthiness in Experiment 2, we observe statistically significant equivalence test results, i.e. the effect of presenting the infographic with versus without icon array on perceived trustworthiness was statistically equivalent to zero. We are thus fairly confident that the effect isn’t large if one existed.

In general, it is true that with a large enough sample one could detect (i.e. find statistically significant differences) even for smaller effects; however, the question is whether that is sensible and meaningful in real world terms.

13. I would find a table helpful showing the main results for both experiments, i.e., the odds ratios (with confidence intervals) for each of the three main comparisons (low vs high certainty in experiment 1 and 2 and low vs no information about certainty in experiment 2) for trustworthiness, perceived effectiveness, and intended behaviour.

We have now included tables with the statistical results of both Experiment 1 and 2 into the manuscript and removed all statistics from the main results reporting. We hope this will be helpful in providing a condensed overview of the results without cluttering the writing. We haven’t made a single table for both experiments, as it would have been quite big and possibly confusing to the reader. We thus chose the route of creating several smaller tables. If you disagree and would like us to put everything into one big table, we would be happy to attempt this. 

Discussion

14. The findings for trustworthiness and perceived effectiveness were very similar for experiment 1 and 2 (as one would expect). How do you explain the apparent difference in findings for intended behaviour?

While perceptions and attitudes in many cases are factors affecting behaviour and thus can be precursors to action, they are qualitatively different. Perceptions of information, such as our measures of trustworthiness of the provided info or people’s perceived effectiveness of the intervention based on the provided info, are qualitatively different from behavioural intentions and behaviour. The Theory of Planned Behavior (Ajzen, 1991), one of the most prominent social psychological theories looking at the factors that bring about human behaviour, postulates that behaviour is a function of intentions which in turn are a function of attitudes, subjective norms, and perceived behavioural control. It might thus be that although people might perceive the information on eye protection as fairly trustworthy, to use the example of our study, that other factors such as their emotional evaluation of carrying out the behaviour (positive or negative feelings about wearing eye protection in public) or normative influences (such as perceptions of what others would think of them if they did) play a role in reducing their intentions to wear eye protection, as to lower the effects observed for the behavioural measure compared to the more perceptual/attitudinal measures. It is thus not unsurprising that we are seeing weaker effects for intended behaviour compared to perceived trustworthiness and effectiveness. Weaker effects are of course also more subject to fluctuating across different samples, such as between our Experiments 1 and 2. We note though that the direction of effects is consistent between behavioural intentions and perceived trustworthiness/effectiveness for both Experiments 1 and 2, with descriptive levels of intentions for the ‘high’ group being higher than levels for the ‘low’ group (Experiment 1), and levels for the ‘no’ group being in between those of the ‘high’ and ‘low’ groups (Experiment 2). An important difference between Experiments 1 and 2 is of course also that we are comparing two groups in Experiment 1 but three groups in Experiment 2.

Ajzen, I. (1991). The theory of planned behavior. Organizational Behavior and Human Decision Processes, 50, 179–211.

15. Although you say that the infographic was “based on empirically-tested good practice recommendations for evidence communication”, I thought there were several problems with it, which you might want to consider in the discussion. First, it does not give any contextual information. The question you asked participants about behaviour was about their “intentions to wear eye protection when in busy public places”. However, all the studies on which the infographic are based were conducted in healthcare settings (except for one small study with no events). There is also no information about how long participants were followed up. The baseline risk (16%) is very high and misleading. It is highly unlikely that anyone not using eye protection in busy public places would experience that high a risk of getting Covid-19. It also is unlikely that random types of eye protection used by ordinary people in crowded public places would have that large of an effect. There is no information about potential harms or downsides. I found the icon array confusing rather than helpful, since I did not understand immediately that each icon represented two people. Also, I am unaware of evidence that indicates that icon arrays effectively communicate information about effects. There is at least one other study that found that an infographic (including an icon array) did not improve understanding (compared to a plain language summary): https://doi.org/10.1016/j.jclinepi.2017.12.003.

We have added some language to illustrate a bit more what we meant by the infographic being ‘based on empirically-tested good practice recommendations for evidence communication’.

The points you bring up are good ones, thank you. We were not setting out to critique the infographic itself, and we agree with your points that the infographic overall might not have been as helpful and ‘good’ as intended by the creators. We have included some language in the Experiment 1 discussion, Experiment 2 methods and discussion to highlight some shortcomings of the icon arrays used in the infographic. We have also included some more references to the relevant icon array literature. We are aware that it is somewhat heterogenous and actually have a paper under review at the moment of our own investigating their usefulness (in the specific instance of communicating an uncertain number) in which we’ve done a more extensive literature survey and conclude that design decisions (such as the choice of denominator and icon) may affect their performance to such a degree as to cause different conclusions to be drawn about their usefulness.

We also acknowledge that we were taking an infographic illustrating estimates taken from a healthcare setting and placing them out of context in a public survey, where the baseline rates would not have been representative. We don’t think that this is likely to have affected our findings, as – even if some people assessed the absolute risks and deemed the infographic less trustworthy on the basis that they seemed too high – these participants were likely to be evenly distributed across our groups and therefore not affecting the comparisons between groups.

More generally, we agree that all the points that you mention that could have enhanced the usefulness of the infographic would be worth testing, e.g. what difference does it make including potential harms or downsides etc. Unfortunately, these aspects are outside the scope of this research, as we merely focused on the effects of quality of evidence communication. We do think though that these are worthy research questions and are the kind of things that we are regularly engaged in investigating. We had set out with our wording mostly to highlight that whilst empirical evidence existed for many elements of the infographic, there appeared to be little empirical basis for the quality of evidence rating. We hope that our new wording makes that clearer.

16. You may want to reconsider or better justify your recommendation that communicators use the word ‘quality’ instead of ‘certainty’ based on participants reporting that it was ‘easier to understand’. I do not agree with your recommendation. First, are you confident that they understood what the word means in this context correctly? I suspect more people are likely to misinterpret the meaning of ‘quality’, given how the word is commonly used, than they are to misinterpret ‘certainty’. Second, whatever term is used, it should always be accompanied by an explanation, since most people are unlikely to understand the basis for judgements about the certainty of the evidence. Third, there is evidence that suggests that people want information about the certainty of the evidence, that they want plain language text – not just numbers, and that using adjectives such as ‘probably’ and ‘may’ can help to communicate the certainty of the evidence (https://doi.org/10.1016/j.jclinepi.2014.04.009, https://doi.org/10.1177/0272989x10375853, https://doi.org/10.1016/j.jclinepi.2019.10.014). I suspect that is more likely to be important for communicators to do (consistently) than it is for them to use ‘quality’ instead of ‘certainty’.

We based our recommendation for using ‘quality’ instead of ‘certainty’ on our empirical finding that people reported finding it easier to understand (an index of both self reported ease and completeness of understanding). It is true that we do not know how people define ‘understanding’ to themselves or for that matter how they understood the meaning of the word in this context, and have amended our wording to reflect that – thank you. We also note that we did not find any significant effects for any of the main outcome measures as detailed in the paper. The only difference we found was for our secondary analysis of understanding. The effect was furthermore fairly small, which we now note explicitly in the paper. Our recommendation for the use of ‘quality’ over ‘certainty’ is therefore not the strongest take away from our work but rather a secondary finding that we think is important to report, albeit with softer wording.

Regarding the second point about an explanation of quality/certainty of evidence; the infographics contained a legend at the bottom that explained the meaning of high/low quality/certainty, although of course we didn’t test participants’ understanding of this.

The infographic did use a plain language explanatory text of the quality/certainty of evidence level below the infographic. It included ‘qualifying’ adjectives such as ‘could be substantially’ or ‘very confident’, e.g. the text for ‘low quality/certainty’ read:

“our confidence in the effect estimate is limited; the true effect could be substantially different from the estimate of the effect”

The text the ‘high quality/certainty’ read:

“we are very confident that the true effect lies close to that of the estimate of the effect.”

This text was the same for both conditions (i.e. the ‘quality’ and the ‘certainty’ wording conditions). 

We would note, though, that in previous research (in the context of direct/quantified uncertainty), we have found that simple adjectives such as ‘about’ or ‘estimated’ do not seem to communicate uncertainty effectively to participants (van der Bles et al 2020 in PLOS). This may be due to participants’ different interpretations of the word ‘uncertainty’, but we tend to caution that numerical ranges appear to be a better way to communicate uncertainty than verbal phrases.

Thank you for sharing the interesting references. They provide important insights into the effects of different presentation formats of different aspects of evidence communication (with the different aspects not being necessarily tested systematically in isolation, e.g. mixing of direct and indirect uncertainty components). We incorporated them into the background section. We’d like to note that in part the focus is slightly different from ours, where we focus on solely quality of evidence information and not so much the presentation format but rather the actual content (i.e. the quality level and its effects). Santesso et al. (2015) for instance showed that people appreciated a plain language summary format that uses the GRADE format to display quality of evidence, i.e. in a table with plus symbol icons and short verbal descriptor. While tremendously useful this study has a different focus to ours, as our work did not test the format in which quality of evidence is presented, rather it tested the effects that the quality of evidence level would have on people’s reactions and perceptions. We tried to make this more clear in the writing.

In general, we agree that there are many elements of good evidence communication that should be considered by communicators. Unfortunately, our studies here could not test all of them.

17. You may want to strengthen your discussion of what you call an ethical dilemma. First, I don’t think public health authorities often do or should present information about effects in isolation. In fact, they frequently fail to provide any quantitative information. So, to the extent there is a dilemma, it is more in relation to including information about the certainty of evidence in connection with a recommendation or advice. In that context, there can be good reasons for recommending something, despite low certainty evidence, and that justification can and should be provided. That has been done by many organisations and people that have made strong recommendations while being transparent about the uncertainty of the evidence for pandemic control measures. Second, many if not most people would consider it unethical to, for example, withhold information about adverse effects of vaccines to persuade people to be vaccinated. How is it different to knowingly withhold information about the certainty of the evidence to persuade people? If you believe there are good reasons to persuade people to wear eye protection in crowded public places, then you should make an honest argument to persuade them. How is it ethically justified to be dishonest and knowingly present untrustworthy information as though it were trustworthy? Third, when there is important uncertainty or disagreement, not being honest and transparent can perpetuate practices that are wasteful and may be harmful. It also can inhibit research to reduce uncertainty and disagreement, and it can undermine trust. Persuasive tactics can infringe on people’s autonomy if information is withheld or if persuasive tactics are not justified. They can also inadvertently harm people. If there is a good justification to persuade people, the basis for doing so should be transparent and persuasive messages should not distort the evidence. This does not mean that clear, actionable messages cannot stand alone. Key messages should be up front, using language that is appropriate for targeted audiences, but it should be easy for those who are interested to dig deeper and find more detailed information, including the justification for a recommendation. When there are important uncertainties, they should be acknowledged.

Thank you for sharing these thoughts and insights. We agree with these excellent points and have included your suggestions into our discussion to make these important points more clear.

Best,

Andy Oxman

We’d like to thank you for your insightful comments, your input has been invaluable. We very much appreciate that you have taken the time to help us improve the manuscript. 

6. PLOS authors have the option to publish the peer review history of their article (what does this mean?). If published, this will include your full peer review and any attached files.

Do you want your identity to be public for this peer review? For information about this choice, including consent withdrawal, please see our Privacy Policy.

Reviewer #1: No

Reviewer #2: No

While revising your submission, please upload your figure files to the Preflight Analysis and Conversion Engine (PACE) digital diagnostic tool, https://pacev2.apexcovantage.com/. PACE helps ensure that figures meet PLOS requirements. To use PACE, you must first register as a user. Registration is free.

Then, login and navigate to the UPLOAD tab, where you will find detailed instructions on how to use the tool. If you encounter any issues or have any questions when using PACE, please email PLOS at figures@plos.org. Please note that Supporting Information files do not need this step.

All figure files were uploaded to PACE and checked against PLOS specifications. The conversions ran without error.

---

## [Decision Letter · Decision Letter 1]

6 Sep 2021

PONE-D-21-13365R1The effects of quality of evidence communication on perception of public health information about COVID-19: two randomised controlled trialsPLOS ONE

Dear Dr. Schneider,

Thank you for submitting your manuscript to PLOS ONE. After careful consideration, we feel that it has merit but does not fully meet PLOS ONE’s publication criteria as it currently stands. Therefore, we invite you to submit a revised version of the manuscript that addresses the points raised during the review process. Please submit your revised manuscript by Oct 21 2021 11:59PM. If you will need more time than this to complete your revisions, please reply to this message or contact the journal office at plosone@plos.org. Please include the following items when submitting your revised manuscript:A rebuttal letter that responds to each point raised by the academic editor and reviewer(s). You should upload this letter as a separate file labeled 'Response to Reviewers'.A marked-up copy of your manuscript that highlights changes made to the original version. You should upload this as a separate file labeled 'Revised Manuscript with Track Changes'.An unmarked version of your revised paper without tracked changes. You should upload this as a separate file labeled 'Manuscript'.If applicable, we recommend that you deposit your laboratory protocols in protocols.io to enhance the reproducibility of your results. Protocols.io assigns your protocol its own identifier (DOI) so that it can be cited independently in the future. For instructions see: https://journals.plos.org/plosone/s/submission-guidelines#loc-laboratory-protocols. Additionally, PLOS ONE offers an option for publishing peer-reviewed Lab Protocol articles, which describe protocols hosted on protocols.io. Read more information on sharing protocols at https://plos.org/protocols?utm_medium=editorial-email&utm_source=authorletters&utm_campaign=protocols.

We look forward to receiving your revised manuscript.

Kind regards,

Jun Tanimoto

Academic Editor

PLOS ONE

Journal Requirements:

Additional Editor Comments (if provided):

Reviewers' comments:

Reviewer's Responses to Questions

**Comments to the Author**

1. If the authors have adequately addressed your comments raised in a previous round of review and you feel that this manuscript is now acceptable for publication, you may indicate that here to bypass the “Comments to the Author” section, enter your conflict of interest statement in the “Confidential to Editor” section, and submit your "Accept" recommendation.

Reviewer #3: All comments have been addressed

2. Is the manuscript technically sound, and do the data support the conclusions?

Reviewer #3: Yes

3. Has the statistical analysis been performed appropriately and rigorously? 

Reviewer #3: Yes

4. Have the authors made all data underlying the findings in their manuscript fully available?

Reviewer #3: Yes

5. Is the manuscript presented in an intelligible fashion and written in standard English?

Reviewer #3: Yes

6. Review Comments to the Author

Reviewer #3: This paper is well written, the topic is interesting, and the results seem correct; the work is acceptable. I make some recommendations for minor revisions. Besides these issues, the work appears methodologically sound and is well written.

##1

Authors should explain details about limitation of their survey and analysis.

##2

Figure captions need to be improved (Author can explain in detail).

##3

Introduction should be improved by adding more recent and related works. For example,

Hypothetical assessment of efficiency, willingness-to-accept and willingness-to-pay for dengue vaccine and treatment: a contingent valuation survey in Bangladesh, Human vaccine and Immunotherapeutics, DOI: 10.1080/21645515.2020.1796424 (2020).

Evolutionary game theory modelling to represent the behavioural dynamics of economic shutdowns and shield immunity in the COVID-19 pandemic. R. Soc. Open Sci. 7: 201095. http://dx.doi.org/10.1098/rsos.201095 (2020).

“Do humans play according to the game theory when facing the social dilemma situation?” A survey study, EVERGREEN, 07(01), 7-14 (2020).

Prosocial behavior of wearing a mask during an epidemic: an evolutionary explanation. Sci Rep 11, 12621 (2021). https://doi.org/10.1038/s41598-021-92094-2.

An evolutionary game modeling to assess the effect of border enforcement measures and socio-economic cost: export-importation epidemic dynamics, Chaos, Solitons & Fractals 146, 110918 (2021).

7. PLOS authors have the option to publish the peer review history of their article (what does this mean?). If published, this will include your full peer review and any attached files.

Reviewer #3: No

---

## [Author Response · Author response to Decision Letter 1]

11 Oct 2021

Response to Reviewers

Reviewers' comments:

Reviewer's Responses to Questions

Comments to the Author

1. If the authors have adequately addressed your comments raised in a previous round of review and you feel that this manuscript is now acceptable for publication, you may indicate that here to bypass the “Comments to the Author” section, enter your conflict of interest statement in the “Confidential to Editor” section, and submit your "Accept" recommendation.

Reviewer #3: All comments have been addressed

2. Is the manuscript technically sound, and do the data support the conclusions?

Reviewer #3: Yes

3. Has the statistical analysis been performed appropriately and rigorously?

Reviewer #3: Yes

4. Have the authors made all data underlying the findings in their manuscript fully available?

Reviewer #3: Yes

5. Is the manuscript presented in an intelligible fashion and written in standard English?

Reviewer #3: Yes

6. Review Comments to the Author

Reviewer #3: This paper is well written, the topic is interesting, and the results seem correct; the work is acceptable. I make some recommendations for minor revisions. Besides these issues, the work appears methodologically sound and is well written.

Thank you for taking the time to review our paper and for the positive feedback.

##1

Authors should explain details about limitation of their survey and analysis.

We have added some more details on study limitations to the discussion.

The revised limitations section now reads:

“This study is limited in that it tested only an online population in the US (albeit quota sampled), and only one health intervention. Further research could broaden this population and context, for example, by using true probability samples, collecting data in multiple countries, engaging in field work, and testing a range of public health interventions. A further limitation is that the quality of evidence information provided in this research was a simple indication of the level, without further details as to the exact reasons for the rating. It would be useful to examine the effects of providing greater nuance and detail on the quality rating, in addition to the effects of adding an explanation for recommendations despite low quality of evidence as outlined above. Furthermore, our work only tested the provision of quality of evidence information in a text format. It is conceivable that providing a quality of evidence label in, for example, a graphical format akin to star ratings, might have a different effect. Lastly, our studies were designed to test overall effects across a broad population. Understanding potential differential effects on different subgroups of the population, such as low and high numeracy individuals, would help to complement knowledge on the effects of quality of evidence communication more broadly, and we thus encourage further research to investigate these relationships more deeply. 

As mentioned in footnote 3, the images shown to participants showed an icon array that had had some of its icons cropped erroneously. This error was consistent across all studies and conditions and hence unlikely to introduce systematic bias that would affect our results in study 1. In study 2, where we tested in addition a difference in presentation format (with and without icon array display), a bias could have been introduced if participants noticed the varying amounts of light grey icons in the two icon array displays and were confused by it. We hence coded the free text responses that participants provided in both studies to identify any comments about the icon arrays. For neither study were there comments relating to confusion about the icon arrays. It therefore seems likely that participants did not notice the error, and we do not expect any influence of it on our observed effects.”

##2

Figure captions need to be improved (Author can explain in detail).

Thank you for suggesting to improve the figure captions. We agree that providing the reader with more details will aid the reader in understanding the figures fast without needing to refer to the methods section. We have elaborated on the text and explanations given in all figure captions.

##3

Introduction should be improved by adding more recent and related works. For example,

Hypothetical assessment of efficiency, willingness-to-accept and willingness-to-pay for dengue vaccine and treatment: a contingent valuation survey in Bangladesh, Human vaccine and Immunotherapeutics, DOI: 10.1080/21645515.2020.1796424 (2020).

Evolutionary game theory modelling to represent the behavioural dynamics of economic shutdowns and shield immunity in the COVID-19 pandemic. R. Soc. Open Sci. 7: 201095. http://dx.doi.org/10.1098/rsos.201095 (2020).

“Do humans play according to the game theory when facing the social dilemma situation?” A survey study, EVERGREEN, 07(01), 7-14 (2020).

Prosocial behavior of wearing a mask during an epidemic: an evolutionary explanation. Sci Rep 11, 12621 (2021). https://doi.org/10.1038/s41598-021-92094-2.

An evolutionary game modeling to assess the effect of border enforcement measures and socio-economic cost: export-importation epidemic dynamics, Chaos, Solitons & Fractals 146, 110918 (2021).

Thank you for the suggestions. Our introduction was initially specifically about studies of quality of evidence communication, and there were no more recent published studies. However, we have taken the reviewer’s suggestion to reframe the introduction to give more background to the specific pandemic situation in which the work was carried out, and agree that this gives the reader more and useful background. This has given us the opportunity to cite some recent and relevant papers on policies and mitigation strategies employed during the pandemic.

7. PLOS authors have the option to publish the peer review history of their article (what does this mean?). If published, this will include your full peer review and any attached files.

Do you want your identity to be public for this peer review? For information about this choice, including consent withdrawal, please see our Privacy Policy.

Reviewer #3: No

---

## [Editor Report · Decision Letter 2]

12 Oct 2021

The effects of quality of evidence communication on perception of public health information about COVID-19: two randomised controlled trials

PONE-D-21-13365R2

Dear Dr. Schneider,

We’re pleased to inform you that your manuscript has been judged scientifically suitable for publication and will be formally accepted for publication once it meets all outstanding technical requirements.

Kind regards,

Jun Tanimoto

Academic Editor

PLOS ONE
---

## [Editor Report · Acceptance letter]

21 Oct 2021

PONE-D-21-13365R2 

The effects of quality of evidence communication on perception of public health information about COVID-19: two randomised controlled trials 

Dear Dr. Schneider:

I'm pleased to inform you that your manuscript has been deemed suitable for publication in PLOS ONE. Congratulations! Your manuscript is now with our production department. 

Kind regards, 

on behalf of

Prof. Jun Tanimoto 

Academic Editor

PLOS ONE